# Drp1 is required for AgRP neuronal activity and feeding

**Sungho Jin[1], Nal Ae Yoon[1], Zhong-Wu Liu[2,3], Jae Eun Song[2,3], Tamas L Horvath[2,3], Jung Dae Kim[1], Sabrina Diano[1,4,5]***

[1]Institute of Human Nutrition, Columbia University Irving Medical Center, New York, United States; [2]Department of Comparative Medicine, Yale University School of Medicine, New Haven, United States; [3]Program in Integrative Cell Signaling and Neurobiology of Metabolism, Yale University School of Medicine, New Haven, United States; [4]Department of Molecular Pharmacology and Therapeutics, Columbia University Irving Medical Center, New York, United States; [5]Department of Cellular and Molecular Physiology, Yale University School of Medicine, New Haven, United States

**Abstract** The hypothalamic orexigenic Agouti-related peptide (AgRP)-expressing neurons are crucial for the regulation of whole-body energy homeostasis. Here, we show that fasting-induced AgRP neuronal activation is associated with dynamin-related peptide 1 (DRP1)-mediated mitochondrial fission and mitochondrial fatty acid utilization in AgRP neurons. In line with this, mice lacking *Dnm1l* in adult AgRP neurons (Drp1 cKO) show decreased fasting- or ghrelin-induced AgRP neuronal activity and feeding and exhibited a significant decrease in body weight, fat mass, and feeding accompanied by a significant increase in energy expenditure. In support of the role for mitochondrial fission and fatty acids oxidation, Drp1 cKO mice showed attenuated palmitic acid-induced mitochondrial respiration. Altogether, our data revealed that mitochondrial dynamics and fatty acids oxidation in hypothalamic AgRP neurons is a critical mechanism for AgRP neuronal function and body-weight regulation.

**\*For correspondence:**
sabrina.diano@columbia.edu

**Competing interests:** The authors declare that no competing interests exist.

## Introduction

The central nervous system (CNS) regulates whole-body energy metabolism through multiple neuronal networks (*Diano, 2013*; *Myers and Olson, 2012*). The hypothalamus has been considered a key area of the brain in regulating metabolism via the ability of hypothalamic neurons to sense, integrate, and respond to fluctuating metabolic signals (*Coll and Yeo, 2013*; *Sandoval et al., 2009*). The hypothalamic arcuate nucleus (ARC) contains two distinct neuronal subpopulations that produce either orexigenic neuropeptides agouti-related peptide (AgRP) and neuropeptide-Y (NPY), or anorexigenic neuropeptides including alpha-melanocyte stimulating hormone (α-MSH) derived from proopiomelanocortin (POMC) (*Batterham et al., 2002*; *Ollmann et al., 1997*; *Roh et al., 2016*). The anatomical location of the hypothalamic ARC allows these neurons to rapidly respond to fluctuations of numerous circulating metabolic signals, including nutrients and hormones (*Gao and Horvath, 2007*). However, the intracellular mechanisms underlying their ability to sense circulating signals, and, specifically nutrients, remain to be elucidated.

Mitochondria are the main powerhouse of the cell by producing adenosine triphosphate (ATP) (*Mattson et al., 2008*; *Picard et al., 2016*). Neurons rely on mitochondrial electron transport chain and oxidative phosphorylation to meet their high energy demands (*Bélanger et al., 2011*). In addition, mitochondria are highly dynamic organelles able to change their morphology and location according to the needs of the cell (*Chan, 2006*). The ability of mitochondria to change their morphological characteristics in response to the metabolic state to match with the needs of the cells occurs

through fusion and fission events, process defined as mitochondrial dynamics. Mitochondrial morphological changes are associated with several proteins, including mitofusin 1 and 2 (MFN1 and MFN2) in the mitochondrial outer membrane and optic atrophy-1 (OPA1) in the mitochondrial inner membrane for mitochondrial fusion (*Kasahara and Scorrano, 2014*; *Youle and van der Bliek, 2012*), whereas mitochondrial fission is regulated by the activity of the dynamin-related protein 1 (DRP1, a mechanochemical protein encoded by the *Dnm1l* gene), which is recruited to the mitochondrial outer membrane to interact with mitochondrial fission factor (Mff) and mitochondrial fission 1 (Fis1) (*Losón et al., 2013*).

Previous studies from our laboratory have shown that NPY/AgRP neuronal activation is associated with changes in mitochondrial morphology and density during fasting or after ghrelin administration (*Andrews et al., 2008*; *Coppola et al., 2007*; *Dietrich et al., 2013*), suggesting that changes in mitochondrial dynamics play a role in the regulation of neuronal activation of these neurons (*Nasrallah and Horvath, 2014*). In addition, we found that high-fat-diet-induced inactivation of NPY/AgRP neurons is associated with mitochondrial dynamics leaning towards mitochondrial fusion in this neuronal population (*Dietrich et al., 2013*). In the present study we interrogated the relevance of mitochondrial fission in AgRP neurons in relation to fuel availability.

## Results

### Fasting induces mitochondrial fission in AgRP neurons

Recent studies have demonstrated that hypothalamic mitochondrial dynamics play a critical role in regulating nutrient sensing (*Dietrich et al., 2013*; *Santoro et al., 2017*; *Schneeberger et al., 2013*; *Toda et al., 2016*). Using electron microscopy, we observed that compared to feeding (0.174 ± 0.007 µm$^2$, p<0.0001; *Figure 1a,c*), fasting resulted in a significant decrease in mitochondrial size (0.130 ± 0.005 µm$^2$, *Figure 1b,c*) in AgRP neurons together with a significant increase in mitochondrial density (0.551 ± 0.032 mitochondria/µm$^2$ of cytosol in fasting vs 0.423 ± 0.026 mitochondria/µm$^2$ of cytosol in feeding; p=0.0031; *Figure 1d*). This was associated with a decrease in mitochondrial aspect ratio (AR; the ratio between the major and minor axis of the ellipse equivalent to the mitochondrion which is indicative of mitochondrial morphological change; 1.629 ± 0.020 in fasting vs 1.769 ± 0.049 in feeding; p=0.0064; *Figure 1e*). However, total mitochondrial coverage in the cytosol (*Figure 1f*) in AgRP neurons was not altered between fed (7.237 ± 0.461% of cytosol) and fasted mice (6.830 ± 0.363% of cytosol; p=0.4853). These observations indicate that food deprivation promotes mitochondrial fission in AgRP neurons, consistent with our prior published work (*Dietrich et al., 2013*).

### Fasting induces significant upregulation of *Dnm1l* mRNA in AgRP neurons

We next performed transcriptomic analysis of AgRP neurons using ribosomal tagging strategy to analyze AgRP neuron-specific mRNA expression levels in fed and fasted conditions using *Agrp*$^{Cre:ERT2}$; RiboTag mice. We observed that compared to fed state (1.036 ± 0.132, n = 5), fasting resulted in a significant increase in *Agrp* mRNA transcript in the input RNAs (5.248 ± 1.176, n = 5; p=0.00741, *Figure 1g*). Compared to the input RNAs, *Agrp* and *Npy* mRNA transcripts were enriched by 27- and 56-fold in the immunoprecipitated RNAs (IP), respectively, in fasted *Agrp*$^{Cre:ERT2}$; RiboTag mice (*Agrp* = input = 5.248 ± 1.176, n = 5; IP = 143.998 ± 22.651, n = 5; p=0.0003, *Figure 1g*; *Npy* = input = 2.680 ± 0.996, n = 5; IP = 151.286 ± 51.683, n = 5; p=0.0207, *Figure 1h*). Conversely, a significant decrease of *Pomc* mRNA transcript in the IP RNAs was found in fed *Agrp*$^{Cre:ERT2}$; RiboTag mice (0.203 ± 0.093, n = 5; p=0.0031, *Figure 1i*) compared to the input RNAs (1.062 ± 0.184, n = 5). Moreover, marginal expression of *Nr5a1*, encoding steroidogenic factor-1 (SF-1, highly restricted to the VMH), was detected in the input (fed = 1.175 ± 0.309, n = 5; fasted = 1.537 ± 0.246, n = 5, *Figure 1j*) and the IP RNAs (fed = 0.846 ± 0.293, n = 5; fasted = 1.826 ± 0.376, n = 5; p=0.074, *Figure 1j*). The *Agrp* IP/input ratio (27.439, *Figure 1g*), *Npy* IP/input ratio (56.45, *Figure 1h*) and enrichment were high, while *Pomc* (IP/input ratio, 0.191, *Figure 1i*) and *Nr5a1*(IP/input ratio, 0.72, *Figure 1i*) were de-enriched, validating the arcuate AgRP neuronal isolation protocol. In support of our mitochondrial morphology data, quantitative real time-PCR (qRT-PCR) analyses revealed that *Dnm1l* mRNA transcript (fed = 0.873 ± 0.337, n = 5;

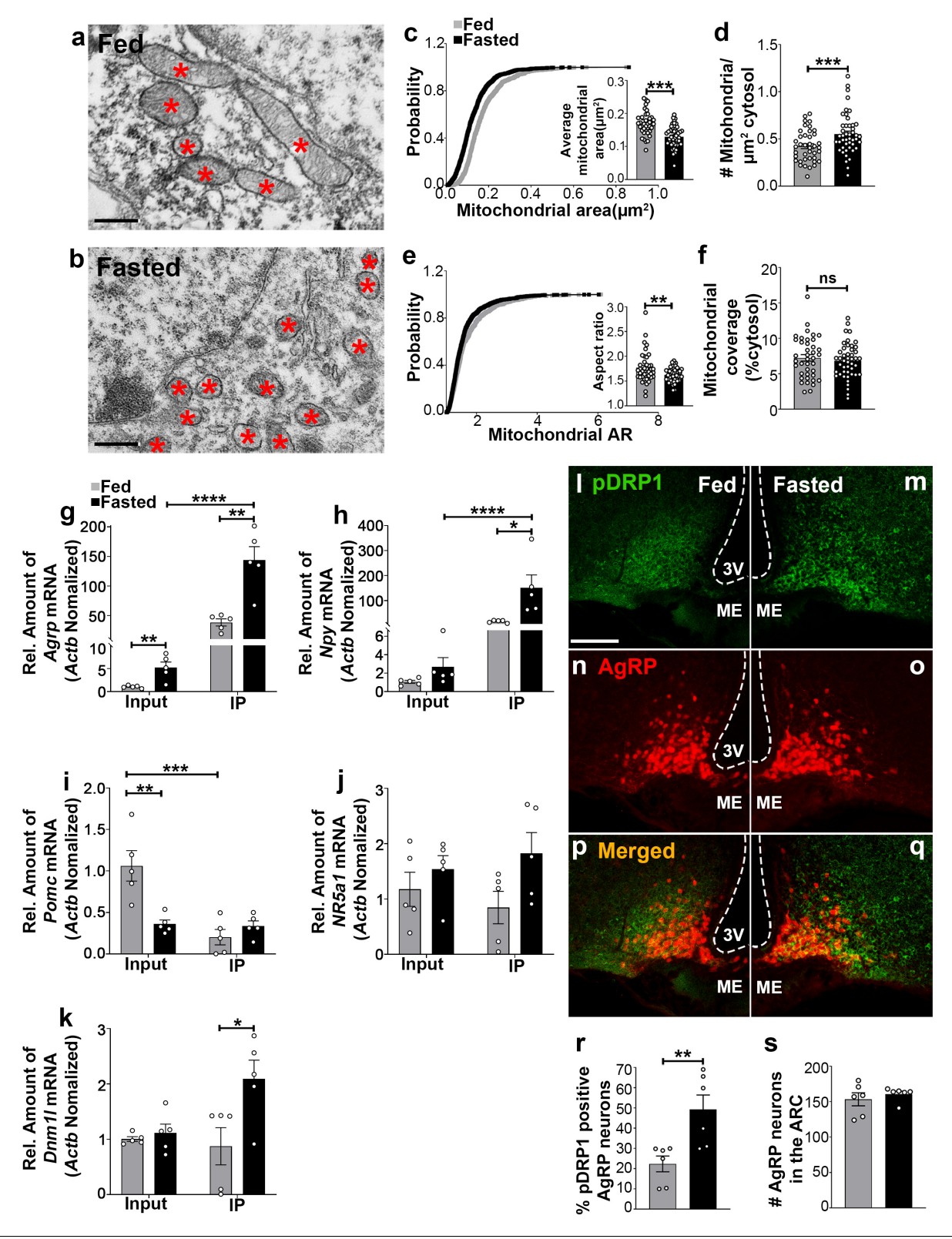

**Figure 1.** Fasting induces mitochondrial fission and activation of DRP1 in AgRP neurons. (a and b) Representative electron micrographs showing mitochondria (asterisks) in an AgRP neuron of 5-month-old fed (a) and the fasted male mouse (b). Scale bar represents 500 nm. (c–f) Cumulative probability distribution of cross-sectional mitochondria area and average mitochondrial area (c), mitochondrial density (d), aspect ratio and a cumulative probability distribution of mitochondrial aspect ratio (e), and mitochondrial coverage (f) in AgRP neurons from fed and fasted male mice (fed mice,

*Figure 1 continued on next page*

Figure 1 continued

n = 779 mitochondria/39 AgRP neurons/4 mice; fasted mice, n = 1559 mitochondria/47 AgRP neurons/6 mice). Data are presented as mean ± SEM. **p<0.01; ***p<0.001 by two-tailed Student's *t*-test. ns = not significant. (g–k) Real-time PCR data showing relative mRNA levels of *Agrp* (g), *Npy* (h), *Pomc* (i), *Nr5a1* (j), and *Dnm1l* (k) in total lysate of hypothalami (Input) and isolated RNA bound to the ribosomes of the hypothalamic AgRP neurons (IP) from 3-month-old fed or fasted mice (n = 5/group). Three animals were pooled for each n. Data are presented as mean ± SEM. *p<0.05; **p<0.01; ***p<0.001; ****p<0.0001 by two-tailed Student's *t*-test. (l–q) Representative micrographs showing immunostaining for phosphorylated DRP1 (at serine 616; pDRP1; green, l and m) and tdTomato (red, representing AgRP, n and o) and merged (p and q) in the hypothalamic ARC of 5-month-old fed and fasted male mice. Scale bar represents 100 µm. 3V = third ventricle; ARC = arcuate nucleus; ME = median eminence. (r) Graph showing the percentage of AgRP neurons immunopositive for pDRP1 (n = 6 mice/group). Data are presented as mean ± SEM. **p<0.01 by two-tailed Student's *t*-test. (s) Graph showing no difference in total AgRP cell number between fed and fasted male mice (n = 6 mice/group). Data are presented as mean ± SEM. p=0.4711 by two-tailed Student's *t*-test.

The online version of this article includes the following source data for figure 1:

**Source data 1.** Fasting induces mitochondrial fission and activation of DRP1 in AgRP neurons.

fasted = 2.095 ± 0.335, n = 5; p=0.0329, *Figure 1k*) was significantly upregulated in AgRP neurons of fasted mice compared to fed mice.

## Fasting induces significant activation of DRP1 protein in AgRP neurons

Mitochondria fission is mediated by DRP1, which is recruited to the outer membrane of mitochondria to promote mitochondrial fragmentation in a GTPase-dependent manner followed by its phosphorylation at serine 616 site (*Liesa et al., 2009*). To examine whether food deprivation is associated with changes in activated Ser616 phosphorylation of DRP1 (pDRP1) levels, we assessed the distribution of pDRP1 immunoreactivity in AgRP neurons in fed and fasted mice. We found that percent of AgRP neurons expressing pDRP1 was significantly increased in fasting (49.2 ± 7.228% of AgRP neurons, n = 6; *Figure 1m,o,q,r*) compared to the fed condition (22.33 ± 3.921% of AgRP neurons, n = 6, p=0.0085, *Figure 1l,n,p,r*). No changes in AgRP cell number were observed between fed (152.2 ± 11.14; n = 5; *Figure 1s*) and fasted mice (160.8 ± 4.028, n = 6; p=0.4527, *Figure 1s*). These data suggest that activation of AgRP neurons in fasting state is closely associated with increased DRP1 activation and, thus, mitochondrial fission, suggesting that DRP1-mediated mitochondrial dynamics may play a role in the regulation of AgRP neuronal activity in fasting state.

## Fasting triggers mitochondrial β-oxidation in the hypothalamic neurons

The hypothalamus is a key region in the control of energy metabolism via the ability of hypothalamic neurons to respond to numerous metabolic signals, including nutrients (*Jin and Diano, 2018*). It has been proposed that hypothalamic availability of free fatty acids controls food intake (*Lam et al., 2005*) and AgRP function (*Andrews et al., 2008*). To investigate the effect of fatty acids on mitochondrial β-oxidation in hypothalamic neurons, we assessed palmitic acid (PA)-induced mitochondrial oxygen consumption rate in primary hypothalamic neuronal cell cultures (*Figure 2*).

First, we analyzed the percentage of tdTomato-expressing AgRP neurons in the cultures and found that about 15% of cells expressed tdTomato (14.75 ± 1.704%; *Figure 2—figure supplement 1a–d*). Next, we assessed the effect of 4-hydroxytamoxifen (to induce tdTomato expression) on neuronal cell viability by trypan blue staining method. Treatment of 2 µM 4-hydroxytamoxifen showed no significant difference in the percentage of cell viability compared to vehicle-treated primary hypothalamic neuronal cultures (*Figure 2—figure supplement 1e*).

A significant difference in PA-induced mitochondrial maximal oxygen consumption rate was observed in primary hypothalamic neurons according to the amount of glucose present in the culture (*Figure 2*). In high glucose concentration, the rate of PA-induced oxygen consumption was significantly lower (*Figure 2a,b*) compared to that measured in low glucose (*Figure 2c,d*). Furthermore, under both high and low glucose conditions, a significant decrease in maximal oxygen consumption rate was observed by the addition of the etomoxir, inhibitor of carnitine palmitoyltransferase-1 (CPT1), transporter of fatty acids into the mitochondria.

Together, these data suggest that similar to fasting state, when glucose levels are low, hypothalamic neurons utilize fatty acids, such as palmitate, as substrates for mitochondrial respiration.

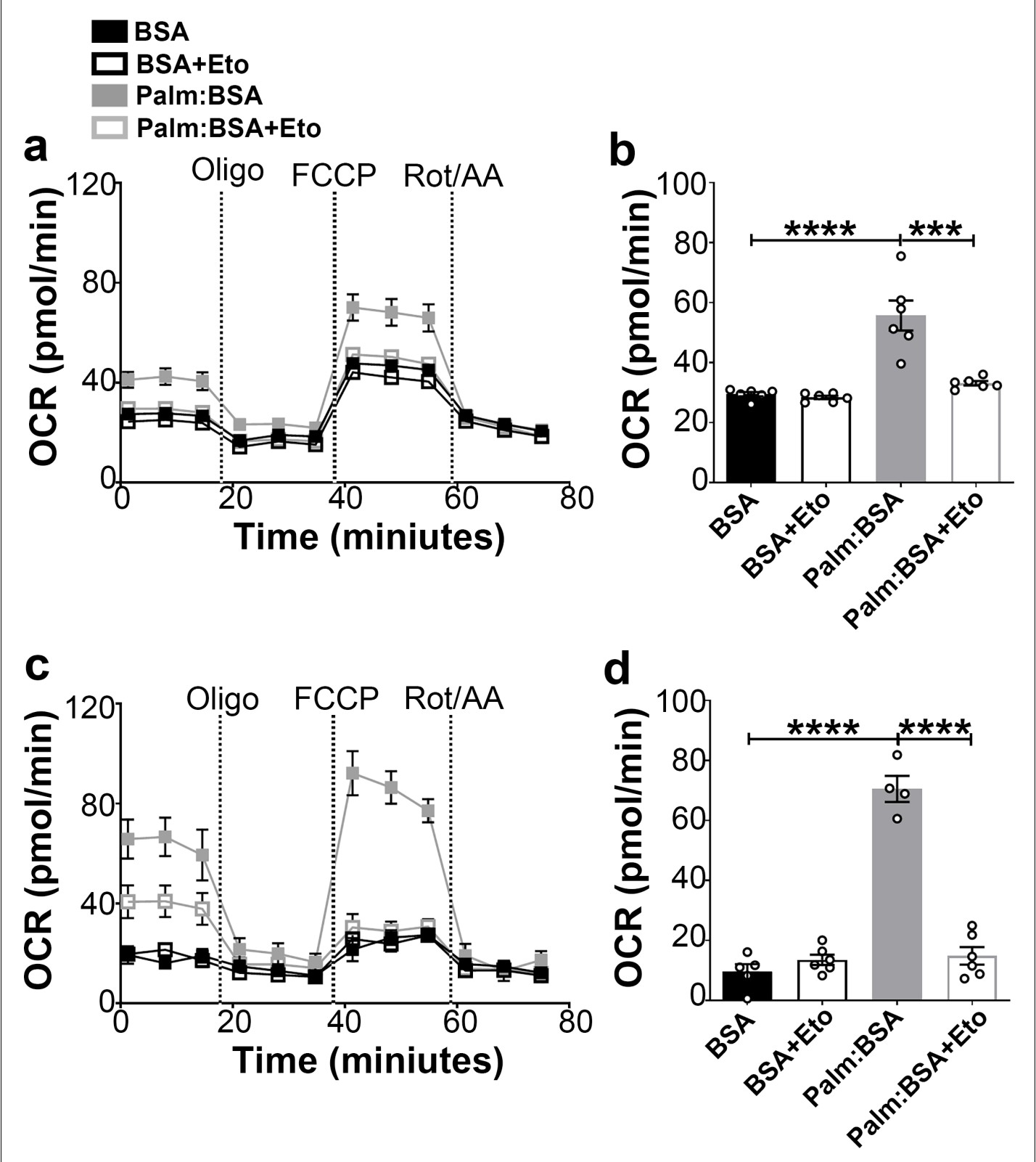

**Figure 2.** Fasting-induced β-oxidation in the hypothalamic neurons. (a) Graphs showing oxygen consumption rate (OCR) under 2.5 mM glucose incubation with or without palmitate-BSA (200 µM) and with or without etomoxir (40 µM) in primary hypothalamic neuronal culture (n = 6–8/group) from $Dnm1l^{+/+}$-$Agrp^{Cre:ERT2}$; tdTomato mice. Cultures were treated with tamoxifen (TMX). (b) Graph showing the quantification of OCR showed in panel (a) in primary hypothalamic neuronal culture. Data are presented as mean ± SEM. *p<0.05; ***p<0.001; ****p<0.0001 by two-way ANOVA with Tukey's post

*Figure 2 continued on next page*

*Figure 2 continued*

hoc analysis for multiple comparisons. (**c**) Graphs showing OCR under low glucose (0.5 mM) with or without palmitate-BSA (200 µM) and with or without etomoxir (40 µM) in primary hypothalamic neuronal culture (n = 6–8/group). (**d**) Graph showing the quantification of OCR shown in panel **c** in primary hypothalamic neuronal culture. Data are presented as mean ± SEM. ***p<0.001; ****p<0.0001 by two-way ANOVA with Tukey's post hoc analysis for multiple comparisons.

The online version of this article includes the following source data and figure supplement(s) for figure 2:

**Source data 1.** Fasting-induced oxidation in the hypothalamic neurons.
**Figure supplement 1.** Percentage of tdTomato-expressing AgRP neurons and cell viability after 4-hydroxytamoxifen treatment in hypothalamic neuronal cell cultures.
**Figure supplement 1—source data 1.** Tomatoes expressing AgRP neurons and cell viability.

## Inducible deletion of *Dnm1l* in AgRP neurons

Next, to investigate the physiological functions of DRP1 in adult AgRP neurons, we generated mice with selective and inducible deletion of *Dnm1l* in AgRP neurons (*Figure 3—figure supplement 1a*). *Agrp^Cre:ERT2*; tdTomato mice (kindly provided by Dr. Joel Elmquist at UTSW; *Wang et al., 2014*) were crossed with *Dnm1l* floxed mice (*Dnm1l^fl/fl*) (*Kageyama et al., 2014*; *Santoro et al., 2017*). As control groups, *Dnm1l^+/+*; *Agrp^Cre:ERT2*; tdTomato mice were injected with tamoxifen and *Dnm1l^fl/fl*; *Agrp^Cre:ERT2*; tdTomato were mice injected with corn oil. *Dnm1l^fl/fl*; *Agrp^Cre:ERT2*; tdTomato mice, referred to here as Drp1 conditional knockout mice (Drp1 cKO mice) were injected with tamoxifen to induce mature-onset deletion of *Dnm1l* in AgRP neurons. To validate our animal model, we analyzed and found limited pDRP1 expression in the AgRP neurons of fasted Drp1 cKO mice (14.15 ± 0.926% of AgRP neurons, n = 4; *Figure 3—figure supplement 1b–h*) compared to fasted *Dnm1l^+/+*; *Agrp^Cre:ERT2*; tdTomato mice (used as control to visualize AgRP neurons; 52.39 ± 3.71% of AgRP neurons, n = 4, p<0.0001, *Figure 3—figure supplement 1b–h*) by immunohistochemistry analysis. No difference in AgRP cell numbers was found between control (150.3 ± 4.423 neurons, n = 4) and Drp1 cKO mice (146 ± 3.082 neurons, n = 4, p=0.4605, *Figure 3—figure supplement 1i*).

## Deletion of *Dnm1l* attenuates fasting-induced mitochondrial fission in AgRP neurons

Next, we analyzed mitochondrial morphological changes in AgRP neurons of Drp1 cKO male mice in fed and fasted states. No differences in mitochondrial size (*Figure 3a–c*), density (*Figure 3d*), aspect ratio (*Figure 3e*), and coverage (*Figure 3f*) were observed between fed and fasted Drp1 cKO male mice, indicating that selective deletion of *Dnm1l* in AgRP neurons prevents fasted-induced mitochondrial fission.

## Deletion of *Dnm1l* in AgRP neurons attenuates mitochondrial functions

First, similar to control-derived cultures, about 15% of cells were tdTomato positive (Drp1 cKO, 14.70 ± 2.901%; *Figure 3—figure supplement 2a–d*). No significant difference was observed between the percentage of tdTomato positive cells in this group (from *Dnm1l^fl/fl*;*Agrp^Cre:ERT2* mice) and the percentage of tdTomato positive cells derived from *Dnm1l^+/+*; *Agrp^Cre:ERT2* mice shown in *Figure 2—figure supplement 1d* (p=0.9883 by two-tailed Student's *t*-test).

We then assessed the effect of 4-hydroxytamoxifen on primary hypothalamic neuronal cell viability. Total viable cell number was measured by trypan blue staining method. Similar to primary hypothalamic neuronal cells isolated from *Dnm1l^+/+*; *Agrp^Cre:ERT2*; tdTomato mice (*Figure 2—figure supplement 1e*), primary hypothalamic neuronal cells isolated from *Dnm1l^fl/fl*; *Agrp^Cre:ERT2*; tdTomato mice showed no significant difference in the percentage of cell viability when treated either with 2 µM 4-hydroxytamoxifen or vehicle (*Figure 3—figure supplement 2e*). Furthermore, a significant reduction of *Dnm1l* mRNA expression was observed in primary hypothalamic neuronal cell cultures treated with 2 µM 4-hydroxytamoxifen derived from *Dnm1l^fl/fl*; *Agrp^Cre:ERT2*; tdTomato mice (0.4512 ± 0.2134, n = 4; *Figure 3—figure supplement 2f*) compared to primary hypothalamic neuronal cell cultures treated with 2 µM 4-hydroxytamoxifen derived from *Dnm1l^+/+*; *Agrp^Cre:ERT2*; tdTomato mice (1.037 ± 0.1825, n = 3, p=0.0493, *Figure 3—figure supplement 2f*).

We then determined PA-induced mitochondrial oxygen consumption rate in primary hypothalamic neuronal cell cultures from Drp1 cKO mice. Contrary to control mice (*Figure 2*), no difference

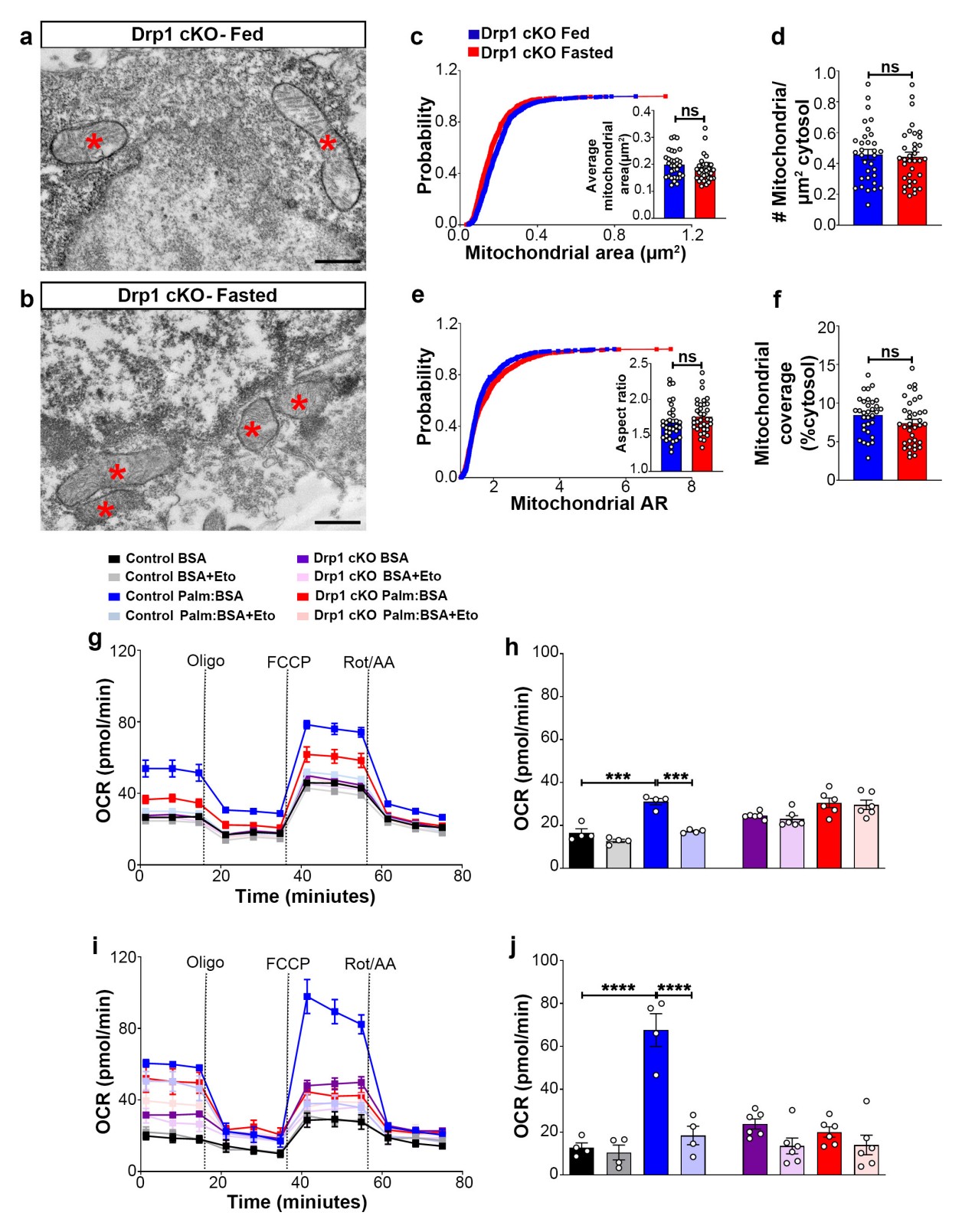

**Figure 3.** Deletion of *Dnm1l* in AgRP neurons affects fasting-induced mitochondrial fission and mitochondrial respiration. (**a and b**) Representative electron micrographs showing mitochondria (asterisks) in an AgRP neuron of the 5-month-old fed Drp1 cKO (**a**) and the fasted Drp1 cKO male mice (**b**). Scale bar represents 500 nm. (**c–f**) Cumulative probability distribution of cross-sectional mitochondria area and average mitochondrial area (**c**), mitochondrial density (**d**), aspect ratio and a cumulative probability distribution of mitochondrial aspect ratio (**e**), and mitochondrial coverage (**f**) in

*Figure 3 continued on next page*

*Figure 3 continued*

AgRP neurons from fed Drp1 cKO (n = 720 mitochondria/32 AgRP neurons/4 mice) and fasted Drp1 cKO male mice (n = 746 mitochondria/35 AgRP neurons/4 mice). Data are presented as mean ± SEM. Two-tailed Student's *t*-test was used for statistical significance. ns = not significant. (g and h) Graphs showing OCR (g) and its quantification (h) under 2.5 mM glucose incubation with or without palmitate-BSA (200 µM) and with or without etomoxir (40 µM) in primary hypothalamic neuronal culture of control (*Dnm1l*^fl/fl^; *Agrp*^Cre:ERT2^; tdTomato treated with vehicle) and Drp1 cKO mice (n = 6–8/group). Data are presented as mean ± SEM. Two-way ANOVA with Tukey's post hoc analysis for multiple comparisons was used for statistical significance. (i and j) Graphs showing OCR (i) and its quantification (j) under low glucose (0.5 mM) incubation with or without palmitate-BSA (200 µM) and with or without etomoxir (40 µM) in primary hypothalamic neuronal culture of control (*Dnm1l*^fl/fl^; *Agrp*^Cre:ERT2^; tdTomato treated with vehicle) and Drp1 cKO mice (n = 6–8/group). Data are presented as mean ± SEM. Two-way ANOVA with Tukey's post hoc analysis for multiple comparisons was used for statistical significance.

The online version of this article includes the following source data and figure supplement(s) for figure 3:

**Source data 1.** Deletion of *Dnm1l* in AgRP neurons affects fasting-induced mitochondrial fission and mitochondrial respiration.

**Figure supplement 1.** Generation of AgRP neurons-specific *Dnm1l* deleted mice.

**Figure supplement 1—source data 1.** Generation of AgRP neuron-specific *Dnm1l*-deleted mice.

**Figure supplement 2.** Percentage of tdTomato-expressing AgRP neurons and cell viability and *Dnm1l* deletion induced by 4-hydroxytamoxifen in hypothalamic neuronal cell cultures.

**Figure supplement 2—source data 1.** Source data for Figure 3—figure supplement 2.

in PA-induced maximal oxygen consumption rate was observed in high (2.5 mM) (*Figure 3g,h*) or low glucose (0.5 mM) (*Figure 3i,j*). In addition, no effects induced by etomoxir incubation were observed in primary hypothalamic neurons derived from Drp1 cKO mice (*Figure 3g–j*), indicating that DRP1 in the hypothalamic AgRP neurons plays an essential role in regulating PA-induced mitochondrial respiration.

## Inducible and selective deletion of *Dnm1l* in AgRP neurons decreases neuronal activation and projection of AgRP neurons in the hypothalamus

To assess the effect of *Dnm1l* deletion on AgRP neuronal activation, we then performed and analyzed immunostaining for Fos in the hypothalamic arcuate nucleus of Drp1 cKO male mice and controls in fasting state (*Figure 4a–f*; *Figure 4—figure supplement 1*). Compared to fasted controls (bregma −1.70 mm: 37.0 ± 2.95% of AgRP neurons, n = 6, *Figure 4a,c,e,g*; bregma −1.46 mm: 40.6 ± 2.71% of AgRP neurons, n = 5, *Figure 4g*, *Figure 4—figure supplement 1a–c*; bregma −2.06 mm: 36.4 ± 3.97% of AgRP neurons, n = 6; total 39.14 ± 2.925% of AgRP neurons, n = 6, *Figure 4g*, *Figure 4—figure supplement 1g–i*), fasted Drp1 cKO male mice showed a significant decrease in immunoreactivity for Fos in AgRP neurons (bregma −1.70 mm: 25.1 ± 3.83% of AgRP neurons, n = 5, p=0.0336, *Figure 4b,d,f,g*; bregma −1.46 mm: 22.6 ± 4.15% of AgRP neurons, n = 4, p=0.0069, *Figure 4g*, *Figure 4—figure supplement 1d–f*; bregma −2.06 mm: 21.6 ± 4.82% of AgRP neurons, n = 4, p=0.0455; total, 24.2 ± 3.58% of AgRP neurons, n = 6, p=0.0097, *Figure 4g*, *Figure 4—figure supplement 1j–l*). No changes in AgRP cell number were observed between control (bregma −1.46 mm, 136.2 ± 10.71 counts, n = 5; bregma −1.70 mm, 142.3 ± 7.89 counts, n = 6; bregma −2.06 mm, 142.7 ± 8.841 counts, n = 6) and Drp1 cKO male mice (bregma −1.46 mm, 147.5 ± 1.50 counts, n = 4, p=0.3858; bregma −1.70 mm, 154.0 ± 3.48 counts, n = 5, p=0.2397; bregma −2.06 mm, 148.0 ± 3.89 counts, n = 4, p=0.6395; total, 150.833 ± 2.753 counts, n = 5, p=0.3467, *Figure 4h*). In agreement with reduced AgRP neuronal activation, a significant reduction in overnight fasting-induced food intake was observed in Drp1 cKO mice (5.155 ± 0.294 g, n = 11, p=0.005) compared to controls (6.391 ± 0.290 g, n = 11, *Figure 4i*).

Furthermore, we analyzed AgRP immunoreactive fibers in one of the major target areas of the hypothalamus, the PVN, of fasted Drp1 cKO male mice and controls. We observed that compared to controls, a significant decrease in the PVN AgRP fluorescent intensity (bregma −0.70 mm: control = 1.000 ± 0.1175, n = 4; Drp1 cKO mice = 0.4372 ± 0.0864, n = 4, p=0.0084, *Figure 5a–c*; bregma −0.82 mm: control = 1.000 ± 0.03906, n = 4; Drp1 cKO mice = 0.3182 ± 0.032222, n = 4, p<0.0001, *Figure 5e–g*; bregma −0.94 mm: control = 1.000 ± 0.1158, n = 4; Drp1 cKO mice = 0.6318 ± 0.06872, n = 4, p=0.0340, *Figure 5i–k*; bregma −1.06 mm: control = 1.000 ± 0.1267, n = 4; Drp1 cKO mice = 0.5641 ± 0.04275, n = 4, p=0.0173, *Figure 5m–o*) and particle number (bregma −0.70 mm: control = 885.25 ± 36.16 counts, n = 4; Drp1 cKO

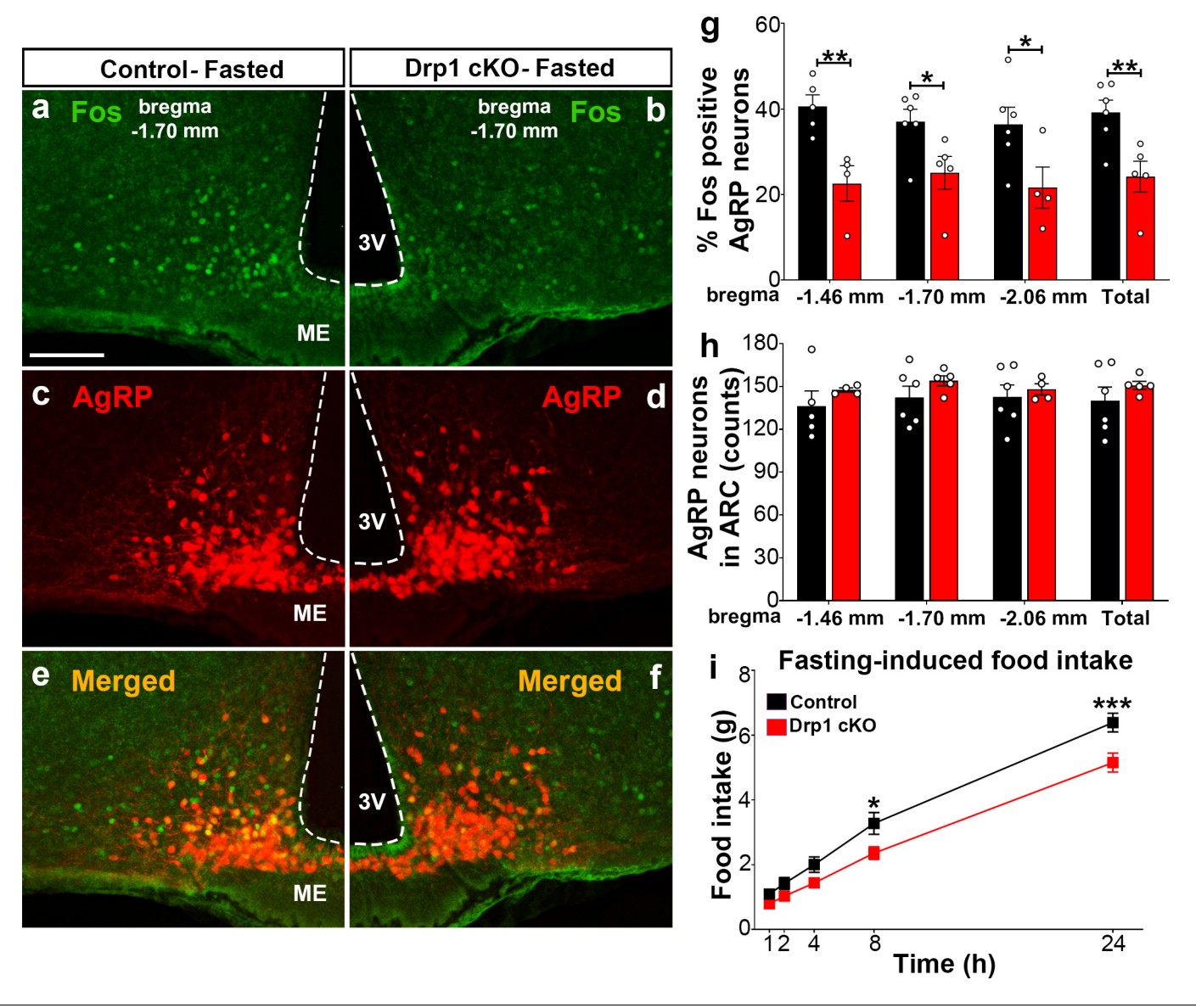

**Figure 4.** *Dnm1l* deficiency in AgRP neurons affects neuronal activation of the hypothalamic AgRP neurons. (**a–f**) Immunostaining for Fos (green, **a and b**) and tdTomato (red, representing AgRP, **c and d**) and merged (**e and f**) in the hypothalamic ARC (bregma −1.70 mm) of a fasted male control (**a, c, and e**) and a Drp1 cKO mouse (**b, d, and f**) at 5 months of age. (**g**) Graph showing the percentage of Fos-positive AgRP neurons in the three bregma coordinates (bregma −1.46 mm, −1.70 mm, and −2.06 mm) corresponding to anterior, medial, and posterior ARC of fasted control (n = 5–6 mice) and Drp1 cKO male mice (n = 4–5 mice) at 5 months of age. Data are presented as mean ± SEM. *p<0.05; **p<0.01 by two-tailed Student's *t*-test. (**h**) Graph showing the number of AgRP neurons in the three bregma coordinates (bregma −1.46 mm, −1.70 mm, and −2.06 mm) corresponding to anterior, medial, and posterior ARC of control (n = 5–6 mice) and Drp1 cKO mice (n = 4–5 mice) at 5 months of age. Data are presented as mean ± SEM. (**i**) Graph showing food intake in male control (n = 11 mice) and Drp1 cKO mice (n = 11 mice) at 4 months of age after overnight fasting (16 hr, 18.00–10.00). Data are presented as mean ± SEM. *p<0.05; ***p<0.001 by two-way ANOVA with Tukey's post hoc analysis for multiple comparisons. 3V = third ventricle; ME = median eminence; ARC = arcuate nucleus.

The online version of this article includes the following source data and figure supplement(s) for figure 4:

**Source data 1.** *Dmn1l* deficiency in AgRP neurons affects neuronal activation of the hypothalmic AgRP neurons.
**Figure supplement 1.** Inducible and selective deletion of *Dnm1l* in AgRP neurons decreases neuronal activation in the ARC.

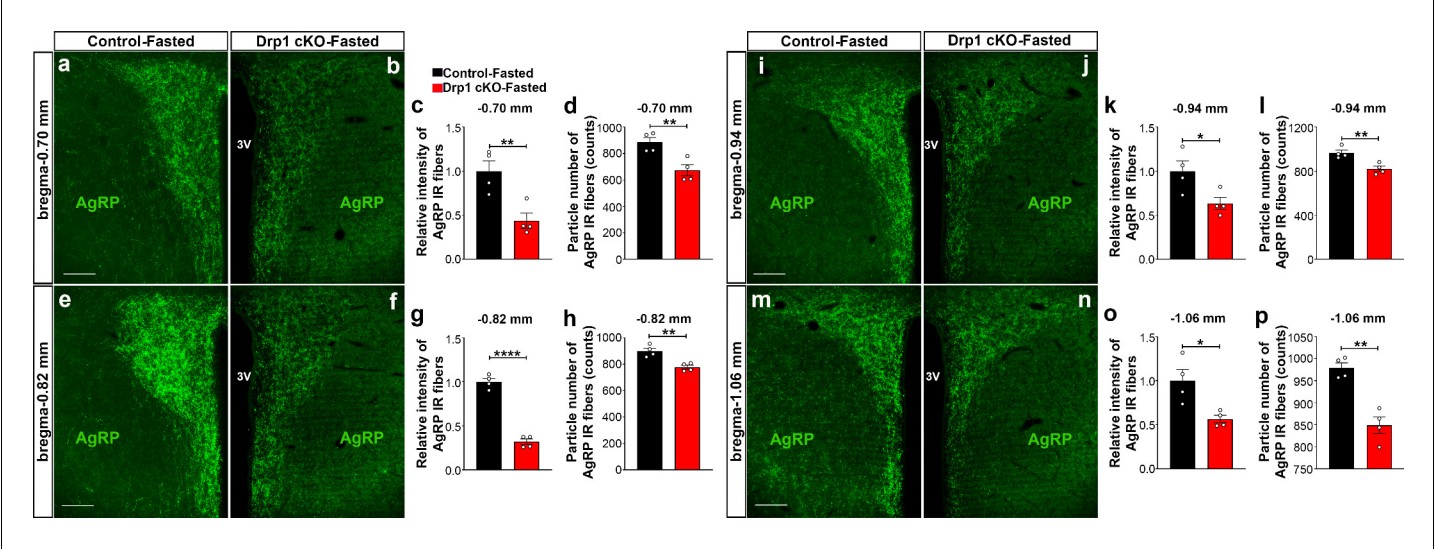

**Figure 5.** AgRP-selective *Dnm1l* deficiency affects AgRP projections within the hypothalamic PVN. (a and b) Immunostaining for AgRP (green) in the PVN (bregma −0.70 mm) of a fasted male control (a) and a fasted Drp1 cKO mouse (b) at 5 months of age. (c and d) Graphs showing the quantification of relative intensity (c) and particle number (d) of AgRP fibers in the PVN (bregma −0.70 mm) of fasted male control and Drp1 cKO male mice (n = 4 mice). (e and f) Immunostaining for AgRP (green) in the PVN (bregma −0.82 mm) of a fasted male control (e) and a fasted Drp1 cKO mouse (f). (g and h) Graphs showing the quantification of relative intensity (g) and particle number (h) of AgRP fibers in the PVN (bregma −0.82 mm) of fasted male control and Drp1 cKO male mice (n = 4 mice). (i and j) Immunostaining for AgRP (green) in the PVN (bregma −0.94 mm) of a fasted male control (i) and a fasted Drp1 cKO mouse (j). (k and l) Graphs showing the quantification of relative intensity (k) and particle number (l) of AgRP fibers in the PVN (bregma −0.94 mm) of fasted male control and Drp1 cKO male mice (n = 4 mice). (m and n) Immunostaining for AgRP in the PVN (bregma −1.06 mm) of a fasted control (m) and a fasted Drp1 cKO mouse (n). (o and p) Graphs showing the quantification of relative intensity (o) and particle number (p) of AgRP fibers in the PVN (bregma −1.06 mm) of fasted control and Drp1 cKO male mice (n = 4 mice). Scale bar represents 100 µm (a, e, i, and m). All data are presented as mean ± SEM. *p<0.05; **p<0.01; ****p<0.0001 by two-tailed Student's *t*-test. 3V = third ventricle; PVN = paraventricular hypothalamus.

The online version of this article includes the following source data and figure supplement(s) for figure 5:

**Source data 1.** AgRP-selective *Dnm1l* deficiency affects AgRP projections within the hypothalmic PVN.

**Figure supplement 1.** Deletion of *Dnm1l* in AgRP neurons affects hypothalamic POMC neurons.

**Figure supplement 1—source data 1.** Deletion of Dnm1l in AgRP neurons affects hypothalamic POMC neurons.

**Figure supplement 2.** Deletion of *Dnm1l* in AgRP neurons does not alter gene expression levels of *Agrp* and *Pomc* in the hypothalamic ARC.

**Figure supplement 2—source data 1.** Source data for Figure 5—figure supplement 2.

mice = 673 ± 42.15, n = 4, p=0.0087, *Figure 5d*; bregma −0.82 mm: control = 899.5 ± 22.15 counts, n = 4; Drp1 cKO mice = 777.6 ± 14.84, n = 4, p=0.0038, *Figure 5h*, bregma −0.94 mm: control = 965.6 ± 25.88 counts, n = 4; Drp1 cKO mice = 821.4 ± 24.35, n = 4, p=0.0067, *Figure 5l*; bregma −1.06 mm: control = 979 ± 11.64 counts, n = 4; Drp1 cKO mice = 848.5 ± 18.79, n = 4, p=0.0010, *Figure 5p*) were observed. Similar results were also observed in the PVN of fasted Drp1 cKO female mice compared to controls (data not shown).

## Deletion of *Dnm1l* in AgRP neurons affects POMC and paraventricular neuronal activation

Next, we analyzed immunostaining for Fos in POMC neurons of Drp1 cKO male mice and their controls (*Figure 5—figure supplement 1*). A significant increase in POMC cells immunoreactive for Fos was observed in Drp1 cKO mice (33.683 ± 2.050% of POMC neurons, n = 4, p=0.0032) compared to controls (22.169 ± 1.297% of POMC neurons, n = 4, *Figure 5—figure supplement 1a–g*). No changes in POMC cell number were observed between control and Drp1 cKO mice (*Figure 5—figure supplement 1h*).

We then analyzed α-MSH fiber immunostaining in the PVN of fasted Drp1 cKO male mice and their controls (*Figure 5—figure supplement 1i,j*). Significant increases in relative intensity (control = 1.000 ± 0.164 counts, n = 4; Drp1 cKO mice = 6.195 ± 0.494, n = 5, p<0.0001, *Figure 5—*

*figure supplement 1k*) and particle number (control = 157.625 ± 13.488 counts, n = 4; Drp1 cKO mice = 393.400 ± 19.290 counts, n = 5, p<0.0001, *Figure 5—figure supplement 1l*) of α-MSH fibers were observed in the PVN of Drp1 cKO mice compared to their controls.

In agreement with a reduced AgRP and an increased POMC neuronal activation, we observed a significant increase in Fos immunopositive cells in the PVN of fasted Drp1 cKO mice (72.600 ± 9.092 counts, n = 5, p=0.0028) compared to their controls (22.750 ± 4.535 counts, n = 4, *Figure 5—figure supplement 1m–o*).

## Deletion of *Dnm1l* in AgRP neurons does not alter gene expression levels of *Agrp* and *Pomc* in the hypothalamic ARC

To assess whether deletion of *Dnm1l* in AgRP neurons had any effect on *Agrp* and *Pomc* mRNA levels in arcuate nucleus, we next performed transcriptomic analysis of AgRP neurons using fasted control and Drp1 cKO Ribotag mice. Quantitative real time-PCR analyses revealed that *Dnm1l* mRNA transcript (fasted control = 4.488 ± 0.685, n = 5; fasted Drp1 cKO = 1.361 ± 0.294, n = 5; p=0.00301, *Figure 5—figure supplement 2a*) was significantly downregulated in AgRP neurons of fasted Drp1 cKO mice compared to fasted control mice, validating the arcuate AgRP neuronal isolation protocol and the mouse model for conditional deletion of *Dnm1l* in AgRP neurons. However, no significant differences in *Agrp* (*Figure 5—figure supplement 2b*) and *Pomc* mRNA levels (*Figure 5—figure supplement 2c*) in the arcuate nucleus were observed between fasted controls and Drp1 cKO mice.

## Deletion of *Dnm1l* in AgRP neurons alters energy metabolism

To determine the physiological outcome of AgRP-specific *Dnm1l* deletion, we assessed the metabolic phenotype of male and female Drp1 cKO mice and their controls. Before starting tamoxifen (TMX) injections at 5 weeks of age, no significant differences in body weight were observed between controls and Drp1 cKO mice in male (body weight = $Dnm1l^{+/+}$; $Agrp^{Cre:ERT2}$-TMX = 18.533 ± 0.390, n = 18; $Dnm1l^{fl/fl}$; $Agrp^{Cre:ERT2}$-TMX = 18.176 ± 0.512, n = 17; $Dnm1l^{fl/fl}$; $Agrp^{Cre:ERT2}$-Corn oil = 18.830 ± 0.308, n = 10; p=0.8649 for $Dnm1l^{+/+}$; $Agrp^{Cre:ERT2}$-TMX versus $Dnm1l^{fl/fl}$; $Agrp^{Cre:ERT2}$-TMX; p=0.9289 for $Dnm1l^{+/+}$; $Agrp^{Cre:ERT2}$-TMX versus $Dnm1l^{fl/fl}$; $Agrp^{Cre:ERT2}$-Corn oil; p=0.7045 for $Dnm1l^{fl/fl}$; $Agrp^{Cre:ERT2}$-TMX versus $Dnm1l^{fl/fl}$; $Agrp^{Cre:ERT2}$-Corn oil, *Figure 6a*).

A significant decrease in body weight of Drp1 cKO male mice compared to controls was observed 3 weeks after the start of TMX treatment (*Figure 6a*) and was maintained through the end of the study when the mice were 20 weeks old (*Figure 6a*; n = 17 per group).

The decrease in body weight of Drp1 cKO male mice was associated with a significant reduction in fat mass (*Figure 6b*; n = 22, p<0.0001) while no significant difference in lean mass was observed (*Figure 6c*; n = 22, p=0.3421) compared to control mice.

Drp1 cKO mice showed significantly lower food intake compared to controls (control = 4.181 ± 0.124 g, n = 14; Drp1 cKO mice = 3.366 ± 0.139 g, n = 18, p<0.0001, total in *Figure 6d,e*), due to a significant reduction of food intake during the dark period (control = 2.937 ± 0.137 g, n = 14; Drp1 cKO mice = 2.471 ± 0.120 g, n = 18, p=0.0236, dark in *Figure 6e*). A significant increase in locomotor activity was observed in Drp1 cKO mice compared to controls (control = 50122.750 ± 5919.799 beam-break counts, n = 14; Drp1 cKO mice = 108400.917 ± 17432.685 beam-break counts, n = 18, p=0.0008, *Figure 6f*), which was due to a significant increase during the dark period (control = 35556.429 ± 4272.846 counts, n = 14; Drp1 cKO mice = 80081.806 ± 13085.949 counts, n = 18, p=0.0136, *Figure 6f*).

The difference in body weight and composition were also associated with significantly increased energy expenditure (control = 303.329 ± 5.760, n = 14; Drp1 cKO mice = 337.096 ± 7.218, n = 18, p<0.0001, *Figure 6g–i*), increased $O_2$ consumption (control = 1245.16 ± 29.94, n = 14; Drp1 cKO mice = 1345.19 ± 31.46, n = 18, p=0.0318, *Figure 6j*) and $CO_2$ production (control = 1140.4 ± 28.18, n = 14; Drp1 cKO mice = 1248.44 ± 27.15, n = 18, p=0.0105, *Figure 6k*) in Drp1 cKO mice compared to control mice, while no significant differences in the respiration exchange rate (RER) were observed between Drp1 cKO mice (0.8734 ± 0.00852, n = 18, p=0.3133) and control mice (0.8855 ± 0.007676, n = 14, *Figure 6l*). Similar to males, female Drp1 cKO mice showed significant differences in body weight and composition, feeding, energy expenditure, $O_2$

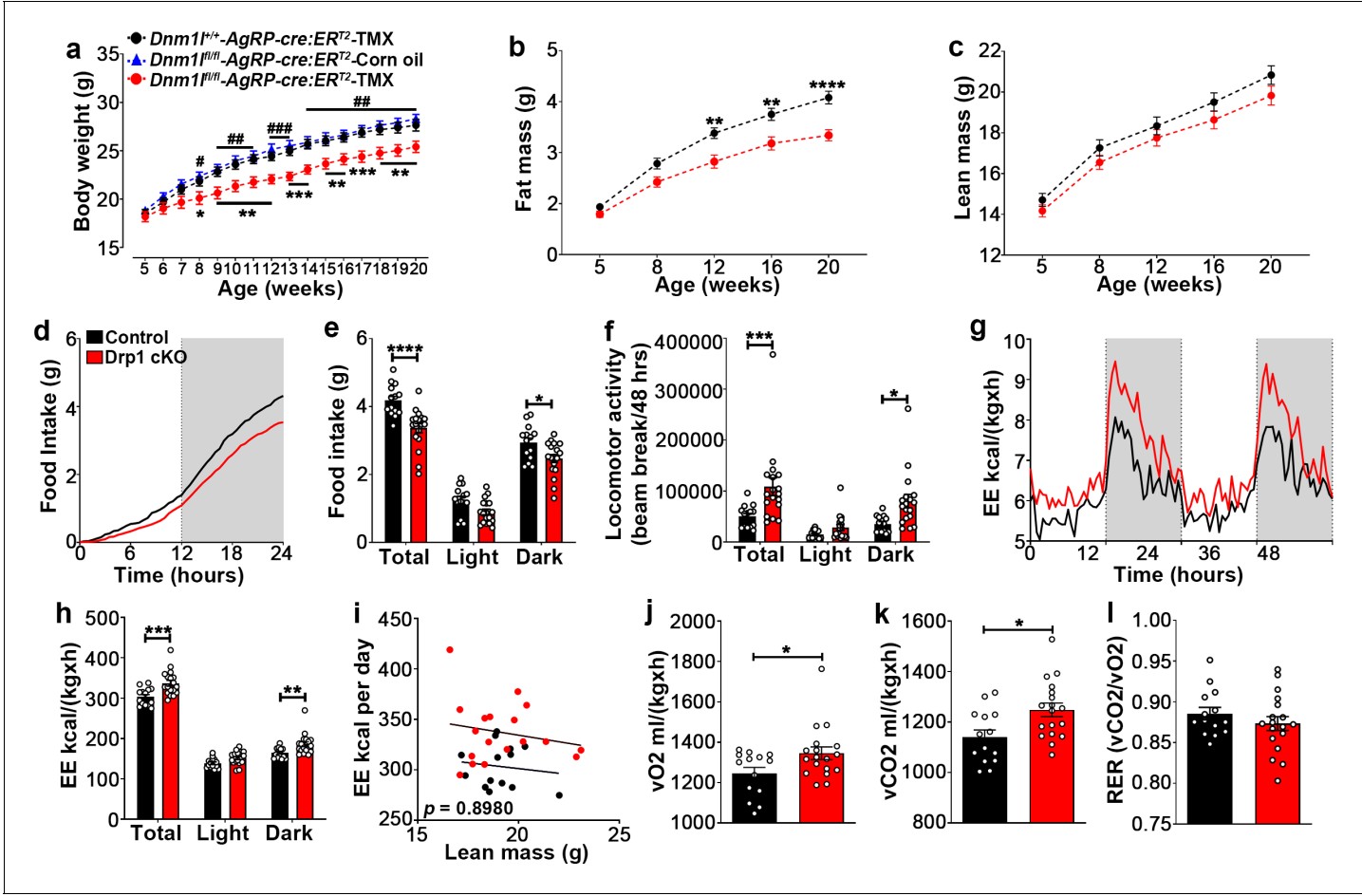

**Figure 6.** Deletion of *Dnm1l* in AgRP neurons affects metabolic phenotype in male mice. (**a**) Graph showing body weight of *Dnm1l*$^{+/+}$; *Agrp*$^{Cre:ERT2}$ mice injected with tamoxifen (n = 18 mice), *Dnm1l*$^{fl/fl}$; *Agrp*$^{Cre:ERT2}$ mice injected with corn oil (n = 10 mice) as control groups, and *Dnm1l*$^{fl/fl}$; *Agrp*$^{Cre:ERT2}$ mice injected with tamoxifen (n = 17 mice). Data are presented as mean ± SEM. *p<0.05; **p<0.01; ***p<0.001 for *Dnm1l*$^{+/+}$; *Agrp*$^{cre:ERT2}$-TMX versus *Dnm1l*$^{fl/fl}$; *Agrp*$^{Cre:ERT2}$-TMX; #p<0.05; ##p<0.01; ###p<0.001 for *Dnm1l*$^{fl/fl}$; *Agrp*$^{Cre:ERT2}$-Corn oil versus *Dnm1l*$^{fl/fl}$; *Agrp*$^{Cre:ERT2}$-TMX by two-way ANOVA with Tukey's post hoc analysis for multiple comparisons. (**b and c**) Graphs showing fat mass (**b**) and lean mass (**c**) of control mice (n = 20 mice) and Drp1 cKO mice (n = 22 mice). Data are presented as mean ± SEM. **p<0.01; ****p<0.0001 by two-way ANOVA with Tukey's post hoc analysis for multiple comparisons. (**d and e**) Graphs showing cumulative 24 hr food intake in 4-month-old control (n = 14) and Drp1 cKO male mice (n = 18) (**d**), and results of food intake as total in the 24 hr cycle and in the dark and light phases of the cycle (**e**; average of 3 days). Gray area represents dark phases. Data are presented as mean ± SEM. *p<0.05; ****p<0.0001 by two-way ANOVA with Tukey's post hoc analysis for multiple comparisons. (**f–l**) Graphs showing locomotor activity (**f**), energy expenditure (**g–i**), consumed O$_2$ (**j**), produced CO$_2$ (**k**), and the respiratory exchange ratio (RER) (**l**) in 4-month-old control (n = 14) and Drp1 cKO male mice (n = 18). Data are presented as mean ± SEM. *p<0.05; ***p<0.001; ****p<0.0001 by two-way ANOVA with Tukey's post hoc analysis for multiple comparisons. p=0.8980 by linear regression analysis (**i**). *p<0.05 by two-tailed Student's *t*-test (**j–k**).

The online version of this article includes the following source data and figure supplement(s) for figure 6:

**Source data 1.** Deletion of *Dnm1l* in AgRP neurons affects metabolic phenotype in male mice.

**Figure supplement 1.** Selective *Dnm1l* deletion in AgRP neurons affects metabolic phenotype in female mice.

**Figure supplement 1—source data 1.** Selective *Dnm1l* deletion in AgRP neurons affects metabolic phenotype in female mice.

**Figure supplement 2.** Deletion of *Dnm1l* in AgRP neurons results in increased BAT and core body temperature.

**Figure supplement 2—source data 1.** Deletion of *Dnm1l* in AgRP neurons results in increased BAT and core body temperature.

consumption, CO$_2$ production, and locomotor activity compared to controls (*Figure 6—figure supplement 1*).

## Deletion of *Dnm1l* in AgRP neurons results in increased brown adipose tissue thermogenesis

BAT thermogenesis is a critical component of the homeostatic energy balance to maintain body temperature (*Morrison and Madden, 2014*). We then examined whether deletion of *Dnm1l* in AgRP neurons affects body temperature. We found that BAT temperature was significantly increased in Drp1 cKO mice compared to control mice (control = 33.388 ± 0.223°C, n = 8; Drp1 cKO mice = 35.038 ± 0.318°C, n = 8, p=0.0008, *Figure 6—figure supplement 2a–c*). Rectal temperature was also significantly increased in Drp1 cKO mice (37.300 ± 0.105°C, n = 8) compared to controls (36.654 ± 0.226°C, n = 8; p=0.0283, *Figure 6—figure supplement 2d*). Similar to males, the rectal temperature of female Drp1 cKO mice (37.181 ± 0.085°C, n = 7) was significantly greater than that of female controls (36.652 ± 0.142°C, n = 9; p=0.0102; *Figure 6—figure supplement 2e*).

## Ghrelin-induced hyperphagia and AgRP activation are dependent on DRP1

Ghrelin, a gut-derived hormone secreted during food deprivation, promotes feeding behavior through NPY/AgRP neurons (*Andrews et al., 2008*). We found a significant decrease in Fos immunoreactivity in AgRP neurons of ghrelin-treated Drp1 cKO mice (30.22 ± 4.652% of AgRP neurons, n = 5, p=0.0001) compared to controls (75.35 ± 3.464% of AgRP neurons, n = 4, *Figure 7a–g*). No difference in the number of AgRP neurons in the ARC was observed between the two groups (*Figure 7h*). In agreement with that, ghrelin-induced hyperphagia was not observed in Drp1 cKO mice compared to controls (*Figure 7i*). Next, we performed patch-clamp whole-cell electrophysiological recordings in slices from Drp1 cKO mice and controls. Consistent with the Fos results, ghrelin significantly increased membrane potential (resting = −46.644 ± 0.502 mV, n = 20; ghrelin = −43.757 ± 0.678 mV, n = 20, p=0.0102; *Figure 7j,l*) and relative firing activity (resting = 100.000 ± 14.584, n = 22; ghrelin = 186.894 ± 20.266, n = 22, p=0.0041; *Figure 7k,l*) of AgRP neurons in control mice, while ghrelin-induced excitation of AgRP neurons was significantly attenuated in Drp1 cKO mice compared to controls (membrane potential, resting = −46.104 ± 0.577 mV, n = 19; ghrelin = −45.677 ± 0.644 mV, n = 19, p=0.9962, *Figure 7j,l*; relative firing activity, resting = 100.000 ± 15.106, n = 20; ghrelin = 119.985 ± 16.502, n = 20, p=0.9644, *Figure 7k,l*). Of note, no differences in the total (*Figure 7—figure supplement 1a*) and active form of ghrelin levels (*Figure 7—figure supplement 1b*) were observed between male control and Drp1 cKO mice in either fed or fasted states. Together, these data suggest that DRP1-mediated mitochondrial fission plays a critical role in regulating ghrelin-triggered AgRP neuronal activity and hyperphagia.

## Discussion

Our findings revealed a crucial role of mitochondrial fission in AgRP neurons in the regulation of hypothalamic feeding control. First, we found that activated AgRP neurons have decreased mitochondrial size accompanied by an increase in mitochondria number suggesting a mitochondrial fission process. In agreement with this, we found that *Dnm1l* mRNA levels and DRP1 activation (*Liesa et al., 2009*) are significantly increased in AgRP neurons of fasted mice compared to fed mice. These data were associated with a significant increase in FA-induced mitochondrial respiration in primary hypothalamic neuronal cells when low glucose levels (similar to fasting) were present compared to higher glucose levels. To determine the physiological relevance of mitochondrial fission in AgRP neurons, we generated a mouse model for conditional deletion of *Dnm1l* in AgRP neurons (Drp1 cKO mice). We found that Drp1 cKO mice, in which fasting did not induce mitochondrial fission and changes in mitochondrial function, had significant decreases in body weight, composition, and feeding that were accompanied by increases in locomotion and energy expenditure. Finally, Drp1 cKO mice also showed attenuated ghrelin-induced hyperphagia and neuronal activity of AgRP neurons. Altogether, these data revealed that DRP1-driven mitochondrial fission in AgRP neurons is an adaptive process enabling these neurons to respond to the changing metabolic environment.

Mitochondria are energy-producing organelles fundamental in support of cellular functions. Mitochondria are highly dynamic organelles able not only to move within the cell to sites where their function is required, but also to fuse (mitochondrial fusion) and divide (mitochondrial fission) in order to maintain proper cellular function.

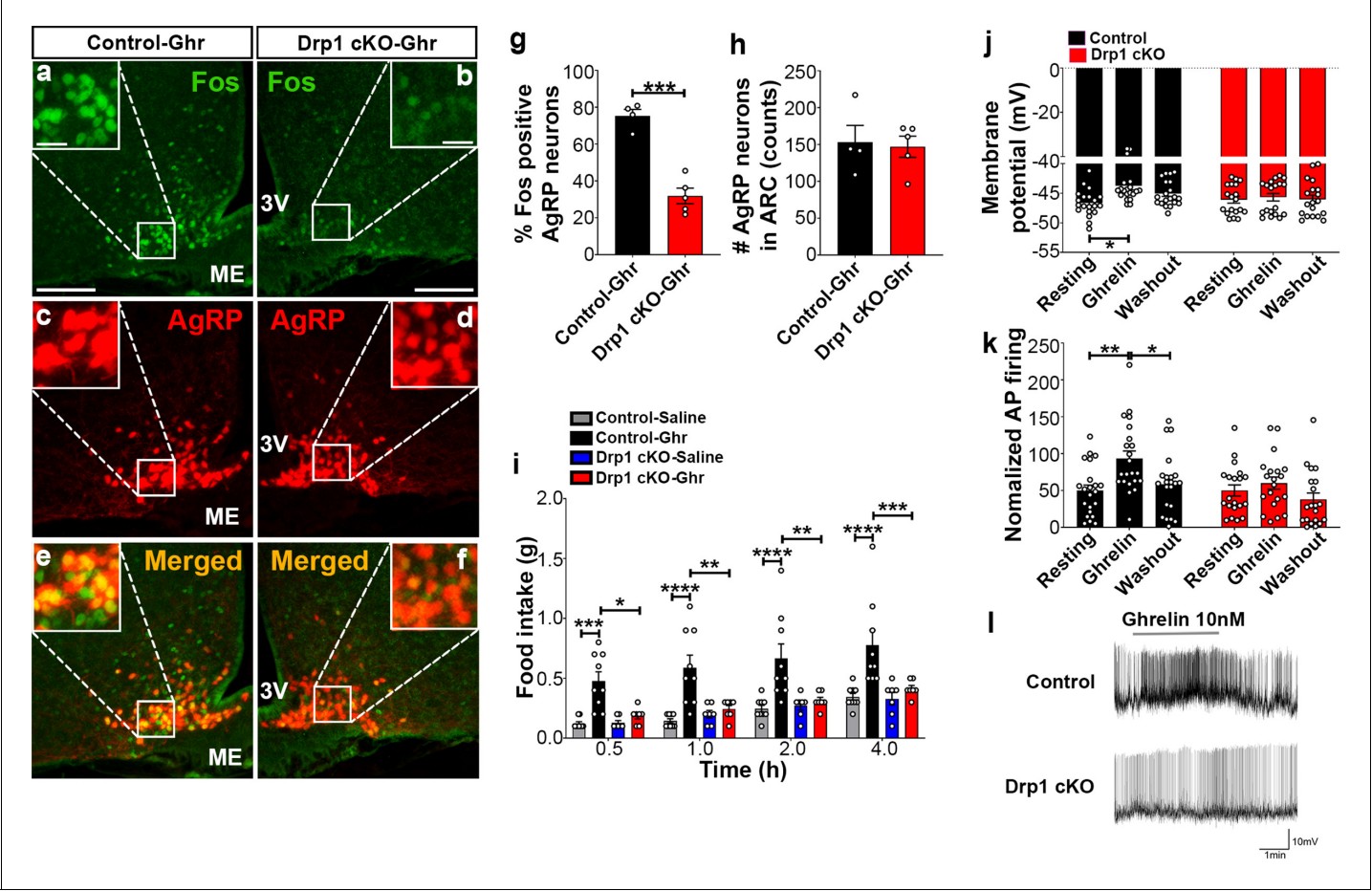

**Figure 7.** Deletion of *Dnm1l* in AgRP neurons attenuates ghrelin-induced neuronal activation and feeding. (**a–f**) Immunostaining for Fos (green, **a** and **b**) and tdTomato (red, representing AgRP, **c** and **d**) and merged (**e** and **f**) in the ARC of a ghrelin-injected male control (**a, c, and e**) and a Drp1 cKO mouse (**b, d, and f**) at 5 months of age. (**g**) Graph showing the percentage of Fos expression in AgRP neurons of ghrelin-injected control and Drp1 cKO mice (n = 4–5 mice). Data are presented as mean ± SEM. ***p<0.001 by two-tailed Student's *t*-test. (**h**) Graph showing the number of AgRP neurons of ghrelin-injected control and Drp1 cKO mice in the hypothalamic ARC (n = 4–5 mice). Data are presented as mean ± SEM. (**i**) Food intake in 4-month-old control and Drp1 cKO female mice at 5 months of age (n = 7–9 mice/group) after either saline or ghrelin injection. *p<0.05; **p<0.01; ***p<0.001; ****p<0.0001; Two-way ANOVA with Tukey's post hoc analysis for multiple comparisons was performed. (**j**) Graph showing the membrane potential in AgRP neurons of 9–11-week-old control (n = 20 cells/10 mice) and Drp1 cKO male mice (n = 19 cells/10 mice) in response to ghrelin. *p<0.05; Two-way ANOVA with Tukey's post hoc analysis for multiple comparisons was performed. (**k**) Graph showing normalized firing rate in AgRP neurons of 9–11-week-old control (n = 20 cells/10 mice) and Drp1 cKO male mice (n = 19 cells/10 mice) in response to ghrelin. Data are presented as mean ± SEM. **p<0.01 for artificial CSF-treated control versus ghrelin-treated control; *p<0.05 for ghrelin-treated control versus washed out control by two-way ANOVA with Tukey's post hoc analysis for multiple comparisons. (**l**) Representative tracers of AgRP neurons from a control and a Drp1 cKO mouse in response to ghrelin. Scale bar represents 100 μm. Scale bar in high magnification image represents 20 μm. 3V = third ventricle; ME = median eminence.

The online version of this article includes the following source data and figure supplement(s) for figure 7:

**Source data 1.** Deletion of *Dnm1l* in AgRP neurons attentuates ghrelin in induced neuronal activation and feeding.

**Figure supplement 1.** Ghrelin levels in male mice.

**Figure supplement 1—source data 1.** Ghrelin levels in male mice.

Mitochondrial fusion and fission are highly regulated processes. Several proteins are involved in these events, including MFN1 and MFN2 and OPA1 for mitochondrial fusion, and Fis1, Mff, and DRP1 for mitochondrial fission (*Pozo Devoto and Falzone, 2017*). Mitochondrial dynamics through fusion and fission processes are also important in maintaining mitochondrial quality control in order to maintain optimal mitochondrial bioenergetic functions (*Twig et al., 2008*). Our data indicate that mitochondrial dynamics and specifically mitochondrial fission play an important role in sensing

changes of nutrients availability in AgRP neurons. First, we observed that incubation of primary hypothalamic neurons with palmitic acid induced a significant increase in mitochondrial respiration when glucose levels were low, mimicking fasting. Fasting induced increased AgRP neuronal activation and increased mitochondrial fission. However, when DRP1-induced mitochondrial fission in AgRP neurons was abolished, palmitic acid-induced mitochondrial respiration was diminished. In association with these, ghrelin-triggered changes in membrane potential and firing frequency of AgRP neurons were significantly attenuated in Drp1 cKO mice, leading to failure in inducing hyperphagia. In line with our results, *Dietrich et al., 2013* have shown that in mice with AgRP-selective deletion of *Mfn1* and *Mfn2*, mediators of mitochondrial fusion process, neuronal firing frequency was impaired in diet-induced obesity mice. The impairment of AgRP neuronal activation was reversed by increasing intracellular ATP levels (*Dietrich et al., 2013*), indicating that the impaired AgRP neuronal firing frequency is likely due to low intracellular ATP levels. In addition to these functions, changes in shape and size of mitochondria may also affect the ability of a cell to distribute its mitochondrial population to specific subcellular locations. This function is especially important in highly polarized cells, such as neurons. Future studies assessing mitochondrial dynamics with changes in mitochondrial subcellular distribution will address this point.

In addition to mitochondria, DRP1 has been also shown to enable peroxisomal fission. Peroxisomes are single-membrane organelles that similar to mitochondria catalyze the breakdown of long chain fatty acids through beta-oxidation and regulate the maintenance of redox homeostasis (*Smith and Aitchison, 2013*). Because of these shared properties and metabolic pathways, we cannot exclude a possible involvement of peroxisomes in the phenotype observed in our mice. Further studies are warranted to address this issue.

Overall, our data unmask that mitochondrial fission in hypothalamic AgRP neurons is a fundamental mechanism that allows these neurons to sense and respond to changes in circulating signals, including hormones such as ghrelin and nutrients such as glucose and palmitic acid, in the regulation of feeding and energy metabolism.

# Materials and methods

**Key resources table**

| Reagent type (species) or resource | Designation | Source or reference | Identifiers | Additional information |
|---|---|---|---|---|
| Strain, strain background (*M. musculus*) | *Agrp*$^{Cre:ERT2}$ | Wang et al., 2013 | | |
| Strain, strain background (*M. musculus*) | *Ai14(Rosa-CAG-LSL-tdTmoato)* | The Jackson Laboratory | Stock No: #007914 | |
| Strain, strain background (*M. musculus*) | *Dnm1l* floxed mouse | *Kageyama et al., 2014* | | |
| Strain, strain background (*M. musculus*) | *Rpl22* floxed mouse | The Jackson Laboratory | Stock No #029977 | |
| Biological sample (*M. musculus*) | Primary hypothalamic neuronal cells | This paper | | Freshly isolated from *M. musculus* in S. Diano Lab. |
| Antibody | Anti-HA antibody (Mouse monoclonal) antibody | Biolegend | Cat# 901513, RRID:AB_2565335 | Immunoprecipitation (5 µl/sample) |
| Antibody | Anti-phosphorylated DRP1 (Ser-616) antibody (Rabbit monoclonal antibody) | Cell Signaling Technology | Cat# 4494, RRID:AB_11178659 | IHC (1:500) |
| Antibody | Anti-Fos antibody (Rabbit polyclonal) antibody | Santa Cruz Biotechnology | Cat# sc-52, RRID:AB_2106783 | IHC (1:2000) |
| Antibody | Anti-POMC (Rabbit polyclonal) antibody | Phoenix Pharmaceuticals | Cat# H-029–30, RRID:AB_2307442 | IHC (1:2000) |
| Antibody | Anti-AgRP (Rabbit polyclonal) antibody | Phoenix Pharmaceuticals | Cat# H-003–57, RRID:AB_2313909 | IHC (1:1000) |

*Continued on next page*

*Continued*

| Reagent type (species) or resource | Designation | Source or reference | Identifiers | Additional information |
|---|---|---|---|---|
| Antibody | Anti-Melanocyte Stimulating Hormone (Sheep polyclonal) antibody | Millipore Sigma | Cat#: ab5087, RRID:AB_91683 | IHC (1:1000) |
| Antibody | Anti-rabbit Alexa Fluor 488 (donkey polyclonal antibody) | Life Technologies | Cat# A21206, RRID:AB_2535792 | IHC (1:500) |
| Antibody | Anti-sheep Alexa Fluor 488 (donkey polyclonal antibody) | Life Technologies | Cat# A11015, RRID:AB_141362 | IHC (1:1000) |
| Antibody | Alexa Fluor 594 anti-goat IgG (donkey polyclonal antibody) | Life Technologies | Cat# A11058, RRID:AB_2534105 | IHC (1:500) |
| Antibody | Biotinylated anti-rabbit IgG (goat polyclonal antibody) | Vector Laboratories | Cat# BA-1000, RRID:AB_2313606 | IHC (1:250) |
| Peptide, recombinant protein | Streptavidin-conjugated Alexa Fluor 488 | Life Technologies | Cat# A21370 | IHC (1:2000) |
| Sequence-based reagent | *Dnm1l* | Thermo Fisher Scientific | Assay ID Mm01342903_m1 | TaqMan Gene Expression Assay (FAM) |
| Sequence-based reagent | *Agrp* | Thermo Fisher Scientific | Assay ID Mm00475829_g1 | TaqMan Gene Expression Assay (FAM) |
| Sequence-based reagent | *Npy* | Thermo Fisher Scientific | Assay ID Mm01410146_m1 | TaqMan Gene Expression Assay (FAM) |
| Sequence-based reagent | *Pomc* | Thermo Fisher Scientific | Assay ID Mm00435874_m1 | TaqMan Gene Expression Assay (FAM) |
| Sequence-based reagent | *Nr5a1* | Thermo Fisher Scientific | Assay ID Mm00446826_m1 | TaqMan Gene Expression Assay (FAM) |
| Sequenced-based reagent | *Actb* | Thermo Fisher Scientific | Assay ID Mm02619580_g1 | TaqMan Gene Expression Assay (FAM) |
| Sequenced-based reagent | *Rn18s* | Thermo Fisher Scientific | Assay ID Mm04277571_s1 | TaqMan Gene Expression Assay (FAM) |
| Peptide, recombinant protein | Ghrelin | ProSpec | Cat# HOR-297-B | |
| Commercial assay or kit | Ghrelin ELISA kit | Millipore Sigma | Cat# EZRGRT-91K | |
| Commercial assay or kit | Rat/Mouse Total Ghrelin ELISA kit | Millipore Sigma | Cat# EZRGRT-90K | |
| Chemical compound, drug | 4-hydroxytamoxifen | Sigma-Aldrich | Cat# H7904 | |
| Chemical compound, drug | Seahorse XF Palmitate-BSA FAO substrate | Agilent Technologies | Cat# 1102720–100 | |
| Chemical compound, drug | Oligomycin | Sigma-Aldrich | Cat# 495455 | |
| Chemical compound, drug | Carbonyl cyanide-p-(trifluoromethoxy) phenylhydrazone | Sigma-Aldrich | Cat# C2920 | |
| Chemical compound, drug | Antimycin A | Sigma-Aldrich | Cat# A8674 | |
| Chemical compound, drug | Rotenone | Sigma-Aldrich | Cat# R8875 | |
| Chemical compound, drug | Avidin–biotin–peroxidase | Vector Laboratories | ABC Elite kit | IHC (1:250) |
| Software, algorithm | FLIR Tools | FLIR Thermal Imaging System | FLIR C2 | |
| Software, algorithm | AxoGraph | AxoGraph Scientific | AxoGraph X | |
| Software, algorithm | KaleidaGraph | Synergy Software | KaleidaGraph v4.5.4 | |
| Software, algorithm | Leading Analysis Software | WaveMetrics | Igor Pro | |

*Continued*

| Reagent type (species) or resource | Designation | Source or reference | Identifiers | Additional information |
|---|---|---|---|---|
| Software, algorithm | Prism software | GraphPad Software | Prism 7.01 software | |
| Other | Standard chow diet | Harlan Teklad | 2018; 18% calories from fat | |
| Other | DAPI | Thermo Fisher Scientific | Cat# P36962 | |

## Animals

All animal care and experimental procedures done in this study were approved by the Yale University (protocol # 10670) and the Columbia University (protocols # AC-AABI0565 and AC-AABH9564) Institutional Animal Care and Use Committees. All mice were housed in a temperature-controlled environment (22–24°C) with a 12 hr light and 12 hr dark (19.00–07.00 hr) photoperiod. Animals were provided standard chow diet (SD) (2018; 18% calories from fat; Harlan Teklad, Madison, WI, USA) and water ad libitum unless otherwise stated. All fasted mice were food deprived for 16 hr (18.00–10.00 hr) prior to the experiment. All mice studied were of the same (mixed) background.

## Generation of experimental mice with inducible deletion of *Dnm1l* specifically in AgRP neurons

We used the inducible Cre/loxP technology to generate mice in which DRP1 was selectively ablated in AgRP neurons (Drp1 cKO mice). First, mice expressing a tamoxifen-inducible Cre recombinase ($CreER^{T2}$) in cells expressing AgRP ($Agrp^{Cre:ERT2}$, Wang et al., 2014) were crossed with Rosa26-lox-stop-lox-tdTomato (*Ai14*; cre-recombinase-dependent expression) mice (Ai14 reporter mice; stock #007914; The Jackson Laboratory, Bar Harbor, ME, USA) to label AgRP-expressing cells. $Agrp^{Cre:ERT2}$; Rosa26-lox-stop-lox-tdTomato ($Agrp^{Cre:ERT2}$; tdTomato) mice have AgRP-expressing cells with the expression of tdTomato by tamoxifen administration. No observation of AgRP-tdTomato expression was found in the absence of tamoxifen administration, indicating that recombination was strictly dependent upon tamoxifen-induced Cre recombinase activation. The mice with $Agrp^{Cre:ERT2}$; tdTomato were then crossed with mice harboring conditional alleles *Dnm1l* floxed ($Dnm1l^{fl/fl}$; Kageyama et al., 2014) to generated mice with inducible deletion of *Dnm1l* specifically in AgRP neurons (Drp1 cKO mice).

$Dnm1l^{fl/fl}$; $Agrp^{Cre:ERT2}$; tdTomato mice injected with corn oil and $Dnm1l^{+/+}$; $Agrp^{Cre:ERT2}$; tdTomato mice injected with tamoxifen (TMX) were used as controls. $Dnm1l^{fl/fl}$; $Agrp^{Cre:ERT2}$; tdTomato mice were injected intraperitoneally (i.p.) with tamoxifen (0.10 mg/g BW for every 3 days with five times fasting) starting at 5 weeks of age to induce mature-onset deletion of *Dnm1l* in AgRP neurons of Drp1 cKO mice, and $Dnm1l^{+/+}$; $Agrp^{Cre:ERT2}$; tdTomato mice were injected with tamoxifen and $Dnm1l^{fl/fl}$; $Agrp^{Cre:ERT2}$; tdTomato were mice injected with corn oil as control groups. Because we found no differences between these two control groups, the majority of the experiments were performed using $Dnm1l^{+/+}$; $Agrp^{Cre:ERT2}$; tdTomato and $Dnm1l^{fl/fl}$; $Agrp^{Cre:ERT2}$; tdTomato mice injected with tamoxifen (to label AgRP neurons with tdTomato expression) as a control and Drp1 cKO group, unless otherwise stated.

## Ribotag assays

We performed transcriptomic profiling by using ribosomal tagging strategy to analyze AgRP neurons-specific mRNA expression in vivo. To avoid the potential disadvantage that the embryonic POMC-expressing progenitor neurons differentiate into AgRP-expressing neurons, we crossed $Agrp^{Cre:ERT2}$ mice (Wang et al., 2014) with *Rpl22* floxed (RiboTag, #029977, The Jackson Laboratories, Bar Harbor, ME, USA) mice to eventually generate $Agrp^{Cre:ERT2}$; RiboTag mice, expressing a hemagglutinin A (HA)-tagged ribosomal protein in the AgRP neurons upon tamoxifen injection. Eleven- to twelve-week-old mice (1 month after the last tamoxifen injection) were used. After mice were anesthetized with isoflurane and decapitated, the brains were rapidly dissected out. To carefully collect the hypothalamic arcuate nucleus (ARC), brain tissues were sectioned in two-millimeter thick coronal sections containing mediobasal hypothalamus (MBH) in a brain matrix. The MBH ARC samples were collected under a stereomicroscope according to the brain atlas for appropriate regions and preventing differences in tissue weight. Three animals were pooled for each N. The MBH ARC samples

from *Agrp*<sup>Cre:ERT2</sup>; RiboTag mice were homogenized by supplemented homogenization buffer (HB-S: 50 mM Tris, pH 7.4, 100 mM KCl, 12 mM MgCl$_2$, and 1 % NP-40 supplemented with 1 mM DTT, 1 mg/ml heparin, 100 µg/ml cycloheximide, 200 U/ml RNasin Ribonuclease inhibitor, and protease inhibitor cocktail). Samples were then centrifuged at 10,000 rpm for 10 min at 4°C. Then, 50 µl of each supernatant was transferred to a new tube serving as input fraction (containing all mRNAs). To isolate polyribosomes, we performed immunoprecipitation of ribosome-bound mRNAs in AgRP neurons. by utilizing anti-HA antibody (5 µl/sample; Cat#901513, Biolegend, San Diego, CA, USA).

RNA was extracted using Qiagen RNeasy Plus Micro Kit (Cat# 74034, Qiagen, Valencia, CA, USA) according to the protocol supplied by the manufacturer. cDNA was synthesized using High Capacity cDNA Reverse transcription Kit (Cat# 4368814, Thermo Fisher Scientific, Waltham, MA, USA). qRT-PCR experiment was performed by Taqman Gene Expression Assay primers (Thermo Fisher Scientific) in triplicates using LightCycler 480 Real-Time PCR System (Roche Diagnostics, Mannheim, Germany). All genes were normalized to *Actb* or *Rn18s*. The 2(-Delta Delta C(t)) method was used to analyze the relative quantification of gene expression. The following primers were utilized: *Dnm1l*, Mm01342903_m1; *Agrp*, Mm00475829_g1; *Npy*, Mm01410146_m1; *Pomc*, Mm00435874_m1; *Nr5a1*, Mm00446826_m1; *Actb*, Mm02619580_g1; *Rn18s*, Mm04277571_s1.

## Metabolic assays

Four-month-old mice were acclimated in metabolic chambers (TSE System-Core Metabolic Phenotyping Center, Yale University) for 3 days before the start of the recordings. Mice were continuously recorded for 2 days, with the following measurements taken every 30 min: food intake, locomotor activity (in the x-, y-, and z-axes), and gas exchange (O$_2$ and CO$_2$; The TSE LabMaster System, Chesterfield, MO, USA). Energy expenditure was calculated according to the manufacturer's guidelines (PhenoMaster Software, TSE System, Chesterfield, MO, USA). The respiratory quotient was estimated by calculating the ratio of CO$_2$ production to O$_2$ consumption. Values were adjusted by body weight to the power of 0.75 (kg$-0.75$) where mentioned. Body composition was measured in vivo by MRI (EchoMRI, Echo Medical Systems, Houston, TX, USA) monthly at 10:00 AM. Body core temperature was measured at 10:00 AM using a thermocouple rectal probe and thermometer (Physitemp instruments, Clifton, NJ, USA). Rectal temperature was measured for repeated three times, and the average was calculated. The temperature of the surface overlying BAT was measured using infrared thermography images (FLIR C2, FLIR Thermal Imaging System, Arlington, VA, USA). The infrared thermography images were taken at least three times and analyzed using FLIR Tools (FLIR Thermal Imaging System, Arlington, VA, USA).

## Phosphorylated-DRP1 immunostaining

Five-month-old mice were deeply anesthetized and transcardially perfused with 0.9% saline containing heparin (10 mg/l), followed by fresh fixative of 4% paraformaldehyde in phosphate buffer (0.1 M PB, pH 7.4) as previously described (*Andrews et al., 2008*; *Diano et al., 2011*; *Toda et al., 2016*). Brains were post-fixed overnight at 4°C and sliced to a thickness of 50 µm using a vibratome (#11000, PELCO easySlicer, TED PELLA Inc, Redding, CA, USA) and coronal brain sections containing the ARC were selected under the stereomicroscope (Stemi DV4, Carl Zeiss Microimaging Inc, Thornwood, NY, USA). After several washes with 0.1 M PB, brain sections were preincubated with 0.2% triton X-100 (Sigma-Aldrich, Saint Louis, MO, USA) and 2% normal goat serum in 0.1 M PB for 30 min to permeabilize tissue and cells. Brain sections were incubated with rabbit anti-phosphorylated-DRP1 (Ser-616) antibody (diluted 1:500 in 0.1 M PB, #4494, Cell Signaling, Technology, Danvers, MA, USA) overnight at room temperature (RT). The following day, brain sections were washed and incubated with a biotinylated goat anti-rabbit IgG (diluted 1:200 in 0.1M PB, BA-1000, Vector Laboratories, Inc, Burlingame, CA, USA) for 2 hr at RT. Sections were then washed and incubated in streptavidin-conjugated Alexa Fluor 488 (diluted 1:2000 in 0.1 M PB, A21370, Life Technologies, Carlsbad, CA, USA) for 2 hr at RT. No staining was performed to visualize AgRP neurons since mice were expressing tdTomato in this neuronal population, which is per se fluorescent. After several washes with 0.1 M PB, brain sections were mounted on glass slides and coverslipped with a drop of Vectashield mounting medium (H-1000, Vector Laboratories, Burlingame, CA, USA). The coverslip was sealed with nail polish to prevent drying and movement under the microscope. All slides were stored in the dark at 4°C.

## Fos immunostaining

Five-month-old mice were deeply anesthetized and transcardially perfused as described above. Immunofluorescent staining was performed using rabbit anti-Fos antibody (diluted 1:2000 in 0.1 M PB, sc-52, Santa Cruz Biotechnology, Dallas, TX, USA) overnight at RT. The following day, brain sections were washed and incubated with a biotinylated goat anti-rabbit IgG secondary antibody (diluted 1:200 in 0.1M PB, BA-1000, Vector Laboratories, Burlingame, CA, USA) for 2 hr at RT. Sections were then washed and incubated in streptavidin-conjugated Alexa Fluor 488 (diluted 1:2000 in 0.1 M PB, A21370, Life Technologies, Carlsbad, CA, USA) for 2 hr at RT. No staining was performed to visualize AgRP neurons since mice were expressing tdTomato in this neuronal population, which is per se fluorescent. For double-label immunohistochemistry of Fos and POMC neurons, sections were processed using goat anti-Fos antibody (diluted 1:2000 in 0.1 M PB, sc-52-G, Santa Cruz Biotechnology, Dallas, TX, USA) overnight at RT. The following day, brain sections were washed and incubated with a Alexa Fluor 594 donkey anti-goat IgG secondary antibody (diluted 1:500 in 0.1M PB, A11058, Life Technologies, Carlsbad, CA, USA) for 2 hr at RT. Brain sections were then incubated with rabbit anti-POMC antibody (diluted 1:2000 in 0.1 M PB, H-029–30, Phoenix Pharmaceuticals, Burlingame, CA, USA). The following day, sections were washed and incubated with Alexa Fluor 488 donkey anti-rabbit IgG secondary antibody (diluted 1:500 in 0.1 M PB, A21206, Life Technologies) for 2 hr at RT. After several washes with 0.1 M PB, brain sections were mounted on glass slides and coverslipped with a drop of vectashield mounting medium (H-1000, Vector Laboratories, Inc, Burlingame, CA, USA) and analyzed with a fluorescence microscope.

## AgRP and α-MSH fiber immunostaining

Five-month-old mice were deeply anesthetized and transcardially perfused as described above. Brain sections containing the hypothalamic paraventricular nucleus (PVN) were selected under the stereomicroscope. Immunofluorescence staining was performed using rabbit anti-AgRP antibody (diluted 1:1000 in 0.1 M PB, H-003–57, Phoenix Pharmaceuticals, Inc) and sheep anti-α-MSH antibody (diluted 1:1000 in 0.1 M PB, ab5087, Millipore Sigma, Burlington, MA, USA) overnight at RT. The following day, brain sections were washed and incubated with anti-rabbit Alexa Fluor 488 (diluted 1:1000 in 0.1M PB, A21206, Life Technologies) and anti-sheep Alexa Fluor 488 (diluted 1:1000 in 0.1M PB, A11015, Life technologies) for 2 hr at RT. After several washes with 0.1 M PB, brain sections were mounted on glass slides, coverslipped with a drop of vectashield mounting medium, and analyzed with a fluorescence microscope.

## Fluorescent image capture and analyses

Five-month-old mice were deeply anesthetized and transcardially perfused as described above. Fluorescent images were captured with Fluorescence Microscope (Model BZ-X710, KEYENCE, Osaka, Japan). For all immunohistochemistry (IHC) analyses, coronal brain sections were anatomically matched (ARC: between −1.46 and −2.06 mm from bregma, PVN: −0.70 and −1.06 mm from bregma) with the mouse brain atlas (*Franklin and Paxinos, 2019*). Both sides of the bilateral brain region (ARC and PVN) were analyzed per mouse. For each mouse, three hypothalamic level-matched per mouse were used to quantify Fos immunoreactive cells in all AgRP and POMC immunostained cells observed in the ARC. The number of immunostained cells was counted manually using ImageJ software (*Schneider et al., 2012*) by an unbiased observer. For area measurements and particle counting, region of interest (ROI) within fluorescence images was manually selected with the mouse brain atlas for ARC, DMH, and PVN, and was then measured by ImageJ software as previously described (*Jin et al., 2016*).

## Hypothalamic primary neuronal cell culture

Eight to ten neonatal (0–1 day old) pups were used for hypothalamic primary neuronal cell culture. For control culture, we used either $Dnm1l^{fl/fl}$; $Agrp^{Cre:ERT2}$; tdTomato mice which neuronal cultures were treated with vehicle (ethanol) or $Dnm1l^{+/+}$; $Agrp^{Cre:ERT2}$; tdTomato mice which neuronal cultures were treated with 4-hydroxytamoxifen (2 μM). Hypothalamic primary neuronal cultures from Drp1 cKO mice ($Dnm1l^{fl/fl}$; $Agrp^{Cre:ERT2}$; tdTomato mice) were treated with 4-hydroxytamoxifen (2 μM). In brief, we carefully removed the MBH of the brain and placed it onto a small culture dish that contains a small volume of Hibernate-A Medium (Cat# A1247501, Gibco-Thermo Fisher Scientific,

Waltham, MA, USA). The tissues dissociated to single cells after digestion with 6 ml of Hibernate-A Medium containing 2.5% of Trypsin-EDTA for 15 min at 37°C. Suspended cells were filtered (40 μm) and centrifuged for 5 min at 1000 rpm and the pellet was re-suspended and plated on XF96 cell culture microplates (Cat# 101085–004, Agilent Technologies, Santa Clara, CA, USA) coated with poly-D-lysine (Cat# P6407, Sigma-Aldrich, Saint Louis, MO, USA) at a density of $1 \times 10^5$ cells per well, and they were cultured in Neurobasal medium (Cat# 21103049, Gibco-Thermo Fisher Scientific, Waltham, MA, USA) supplemented with 1% penicillin–streptomycin, 2% B-27 Supplement (Cat# 17504044, Gibco-Thermo Fisher Scientific, Waltham, MA, USA), and GlutaMAX-I (Cat# 35050061, Gibco-Thermo Fisher Scientific, Waltham, MA, USA). After 10 days in culture, primary neuronal cells isolated from control (*Dnm1l*[+/+]; *Agrp*[Cre:ERT2]; tdTomato) and Drp1 cKO mice were treated with 2 μM 4-hydroxytamoxifen (H7904, Sigma-Aldrich, Saint Louis, MO, USA) for expression of a CreER recombinase while the other control group (generated from *Dnm1l*[fl/fl]; *Agrp*[Cre:ERT2]; tdTomato mice) was treated with vehicle (ethanol) to assess the effect of 4-hydroxytamoxifen on cell viability. Primary neuronal cells were used for the measurement of mitochondria fatty acid oxidation 5 days later.

## Cell quantification in cultures

Cells were analyzed by capturing six to eight random fields per coverslip. For the quantitative analysis of cell number, tomato expressing cells in DAPI (Cat# p36962, Thermo Fisher Scientific, Waltham, MA, USA)-stained cultures were manually counted using Image J software. Cells were visualized using Fluorescence Microscope (Model BZ-X710, KEYENCE, Osaka, Japan). Five coverslips per group were counted within an experiment.

## Viability assay in cultures

Neuronal cell viability was determined by trypan blue exclusion assay in cultures maintained in each condition. The cultures were stained with 0.4% trypan blue (Cat# 302643, Sigma-Aldrich, Saint Louis, MO, USA) for 15 min at room temperature and then washed with phosphate-buffered saline (PBS). And then, 10 μL of suspended cells was loaded into each chamber of the hemocytometer. Counts were performed by triplicate by one analyst under a 40× objective according to the standard methodology. The non-stained (live) and Trypan blue-stained (dead) cell counts were counted and calculated in three randomly selected areas (0.2 mm$^2$) in each well (n = 5 per treatment condition) to calculate the cell viability percentage.

## Measurement of mitochondrial fatty acid oxidation assay

The fatty acid oxidation (FAO) was measured using a microfluorimetric Seahorse XF96 Analyzer (Agilent Technologies, Santa Clara, CA, USA) according to the protocol supplied by the manufacturer with minor modifications. Cells were starved with minimal substrate neurobasal-A medium (Cat# 10888022, Thermo Fisher Scientific) for 24 hr. The minimal substrate medium included 1% B-27 Supplement (Cat# 17504044, Gibco-Thermo Fisher Scientific, Waltham, MA, USA), 1 mM glutamine, 0.5 mM carnitine, and 2.5 or 0.5 mM of glucose. The day of the assay, 45 min prior to the assay, starved cells were washed and incubated with Seahorse XF Base medium Minimal DMEM (Cat# 102353–100, Agilent Technologies, Santa Clara, CA, USA) supplemented with 2.5 or 0.5 mM glucose and 0.5 mM carnitine in a non-CO$_2$ 37°C incubator. Fifteen minutes prior to the assay, 40 μM etomoxir was added to the cells to measure endogenous fatty acid uptake for FAO. Palmitate-BSA or BSA control (Seahorse XF Palmitate-BSA FAO substrate, Cat# 1102720–100, Agilent Technologies, Santa Clara, CA, USA) were added to cells right before initiating the XF assay. During the assay, cells were exposed to compounds in the following order: 5 μM of oligomycin (Cat# 495455, Sigma), 10 μM of FCCP [carbonyl cyanide-p-(trifluoromethoxy) phenylhydrazone] (Cat# C2920, Sigma-Aldrich, Saint Louis, MO, USA), 10 μM of antimycin A (Cat# A8674, Sigma-Aldrich, Saint Louis, MO, USA), and 5 μM of rotenone (Cat# R8875, Sigma-Aldrich, Saint Louis, MO, USA). Wave 2.6.0 (Agilent Technologies software, Santa Clara, CA, USA) software was used to analyze the parameters.

## Electrophysiology analysis

Electrophysiology analyses were performed as previously described (*Toda et al., 2016*). Briefly, 11–12-week-old mice were used for recordings. After mice were anesthetized with isoflurane and decapitated, the brains were rapidly removed and immersed in an oxygenated cutting solution at 4°

C containing (in mM): sucrose 220, KCl 2.5, $NaH_2PO_4$ 1.23, $NaHCO_3$ 26, $CaCl_2$ 1, $MgCl_2$ 6, and glucose 10, pH (7.3) with NaOH. After being amputated to a small tissue block, coronal slices containing the hypothalamus (300 μm thick) were cut with a vibratome. After preparation, slices were stored in a holding chamber with an oxygenated (with 5% $CO_2$% and 95% $O_2$) artificial cerebrospinal fluid (aCSF) containing (in mM): NaCl 124, KCl 3, $CaCl_2$ 2, $MgCl_2$ 2, $NaH_2PO_4$ 1.23, $NaHCO_3$ 26, glucose 3, pH 7.4 with NaOH. The slices were eventually transferred to a recording chamber perfused continuously with aCSF at 33°C at a rate of 2 ml/min after at least a 1 hr recovery in the storage chamber. Perforated patch recording was performed in AgRP-Tomato neurons of the ARC under voltage and current clamp. The membrane and spontaneous action potential were recorded in AgRP neurons under zero current clamp condition. For ghrelin-induced AgRP neuronal activation, baseline activity was recorded for at least 15 min. Slices were then perfused with 10 nM ghrelin, diluted in aCSF for 3 min, followed by a washout (with no ghrelin). At the end of the perforated patch recordings, the membrane of every cell was ruptured and whole-cell patch recording measured to check current–voltage relationship. All data were sampled at 5 kHz, filtered at 2.4 kHz, and analyzed with an Apple Macintosh computer using AxoGraph X (AxoGraph Scientific, Foster City, CA, USA). Statistics and plotting were performed with KaleidaGraph (Synergy Software, Inc, Reading, PA, USA) and Igor Pro (WaveMetrics, Lake Oswego, OR, USA). The average firing rate was calculated in the last 2 min of each control period or treatment application. All the experiments were performed blindly to the electrophysiologist.

## Ghrelin administration

Individually housed 4-month-old mice were i.p. injected with either 0.9% saline (#0409-1966-12, Hospira Inc, Lake Forest, IL, USA) or ghrelin (10 nmol, HOR-297-B, ProSpec, Rehovot, Israel) at 9:00 AM. Immediately after injection, mice were returned to their home cages, which contained a pre-weighed amount of food. The remaining food was measured at 0.5, 1, 2, and 4 hr post-injection. For immunostaining, mice were injected with ghrelin at 9:00 AM and 1 hr later, mice were deeply anesthetized and transcardially perfused, and brains were dissected and sectioned (50 μm) using a vibratome. Brain sections were processed for Fos immunostaining. Fluorescent images were captured with a Fluorescence Microscope (BZ-X710, KEYENCE, Osaka, Japan). Fos/AgRP positive cells were counted using ImageJ software.

## Electron microscopy analysis

Mice (5 months old) were deeply anesthetized and transcardially perfused with 0.9% saline containing heparin (10 mg/l), followed by fresh fixative (4% paraformaldehyde, 15% picric acid, 0.1% glutaraldehyde in 0.1 M PB). Brain coronal sections were immunostained with rabbit anti-RFP antibody (diluted 1:1000 in 0.1 M PB, 600-401-379, Rockland Immunochemicals, Limerick, PA, USA) for AgRP neurons. After several washes with 0.1 M PB, sections were incubated with biotinylated goat anti-rabbit IgG (diluted 1:250 in 0.1 M PB, BA-1000, Vector Laboratories, Burlingame, CA, USA) for 2 hr at RT, and then rinsed in 0.1 M PB three times 10 min each time and incubated for 2 hr at RT with avidin–biotin–peroxidase (ABC; diluted 1:250 in 0.1 M PB; ABC Elite kit, Vector Laboratories). The immunoreaction was visualized with 3,3-diaminobenzidine (DAB). Sections were then osmicated (1% osmium tetroxide) for 30 min, dehydrated through increasing ethanol concentrations (using 1% uranyl acetate in the 70% ethanol for 30 min), and flat-embedded in araldite between liquid release-coated slides (Electron Microscopy Sciences, Hatfield, PA, USA). After capsule embedding, blocks were trimmed. Ribbons of serial ultrathin sections were collected on Formvar-coated single slot grids and examined using a Philips CM-10 electron microscope. Mitochondria morphology in AgRP neurons of fed and fasted mice were analyzed using ImageJ software as previously described (*Toda et al., 2016*).

## Measurement of circulating hormones

Five-month-old mice were deeply anesthetized and decapitated. The blood was collected into a capillary tube (Microvette, CB 300 Z, Sarstedt, Nümbrecht, Germany) containing 0.2 mg 4-(2-aminoethyl)-benzene-sulfonyl fluoride (AEBSF, Roche, Basel, Switzerland). Serum from blood samples was obtained by centrifugation at 3000 rpm for 15 min, and each circulating hormone was determined using a commercially available ELISA kit for total ghrelin (Rat/Mouse Total Ghrelin ELISA kit,

EZRGRT-91K, Millipore Sigma, Burlington, MA, USA) and active ghrelin (Rat/Mouse Total Ghrelin ELISA kit, EZRGRT-90K, Millipore Sigma, Burlington, MA, USA). Serum samples and standards were analyzed in duplicate. All procedures were performed by following the manufacturer's protocol.

## Statistical analysis

Two-way ANOVA was used to determine the effect of the genotype and treatment with the Prism 7.01 software (GraphPad Software). For repeated measures analysis, ANOVA was used when values over different times were analyzed. When only two groups were analyzed, statistical significance was determined by an unpaired Student's $t$-test. A value of $p<0.05$ was considered statistically significant. All data is shown as mean ± SEM, unless otherwise stated.

## Acknowledgements

This work was supported by NIH R01 DK097566, DK107293, and DK120321 to SD.

## Additional information

### Funding

| Funder | Grant reference number | Author |
|---|---|---|
| National Institute of Diabetes and Digestive and Kidney Diseases | DK097566 | Sabrina Diano |
| National Institute of Diabetes and Digestive and Kidney Diseases | DK107293 | Sabrina Diano |
| National Institute of Diabetes and Digestive and Kidney Diseases | DK120321 | Sabrina Diano |
| National Institutes of Health | NIH R01 AG052005 | Tamas L Horvath |
| National Institutes of Health | NIH R01 DK126447 | Tamas L Horvath |

The funders had no role in study design, data collection and interpretation, or the decision to submit the work for publication.

### Author contributions

Sungho Jin, Data curation, Formal analysis, Investigation, Methodology, Writing - original draft; Nal Ae Yoon, Data curation, Formal analysis, Investigation, Methodology; Zhong-Wu Liu, Formal analysis, Investigation; Jae Eun Song, Methodology; Tamas L Horvath, Resources; Jung Dae Kim, Investigation, Methodology; Sabrina Diano, Conceptualization, Formal analysis, Supervision, Funding acquisition, Writing - original draft, Project administration, Writing - review and editing

### Author ORCIDs

Sungho Jin ⓘ https://orcid.org/0000-0002-5915-4042
Tamas L Horvath ⓘ http://orcid.org/0000-0002-7522-4602
Sabrina Diano ⓘ https://orcid.org/0000-0002-7921-2617

### Ethics

Animal experimentation: All animal work was approved by the Institutional Animal Care and Committee of Columbia University (protocols # protocols # AC-AABI0565 and AC-AABH9564) and Yale University (protocol #10670).

### Decision letter and Author response

Decision letter https://doi.org/10.7554/eLife.64351.sa1
Author response https://doi.org/10.7554/eLife.64351.sa2

## Additional files

### Supplementary files
• Transparent reporting form

### Data availability
All data generated or analyzed during this study are included in the manuscript and supporting files.

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
