## [Decision Letter]

**Acceptance summary:**

The importance of mitochondrial fission for normal physiological function was established in this paper. The authors selectively inactivated a key protein required for mitochondrial fission in AgRP neurons that are important for feeding and protection from starvation. This manipulation resulted in reduced activity of these neurons and a lean phenotype, clearly demonstrating that disruption of mitochondrial dynamics impacts the normal functions of these neurons.

**Decision letter after peer review:**

Thank you for submitting your article "DRP1 is required for AgRP neuronal activity and feeding" for consideration by *eLife*. Your article has been reviewed by three peer reviewers, including Richard D Palmiter as the Reviewing Editor and Reviewer #1, and the evaluation has been overseen by Matt Kaeberlein as the Senior Editor.

The reviewers have discussed the reviews with one another and the Reviewing Editor has drafted this decision to help you prepare a revised submission.

The authors of this paper demonstrate that mitochondrial fission is important for normal AgRP neuron function. Mice lacking DRP1, a key enzyme mediating fission, in AgRP neurons have lower neuronal activity, respond less well to ghrelin, gain less weight than control mice and have reduced fatty acid metabolism. The authors provide a compelling set of data to substantiate their claims and describe their results clearly; however, a few issues need to be addressed.

Essential revisions:

1) The authors should provide more compelling evidence that energy expenditure (EE) is increased in mice lacking Drp1 in AgRP neurons because measuring EE per gram body weight when the weights of the mice differs can be misleading.

2) For the data in Figure 2, it is not clear to what extent AgRP neurons contribute to the difference fatty acid oxidation. Comparison of the data in Figure 2 (cultures from wild-type mice) with the data in Figure 3 (Drp1 cKO mice) are used to demonstrate the importance of Drp1 for fatty acid oxidation. This comparison is troublesome because they were done with primary cultures prepared and assayed at different times and the extent of Drp1 KO was not established. A better design would be to prepare cultures from the *Agrp-CreERt::Dnml1(fl/fl)* mice that were either treated with tamoxifen or not and then assayed for fatty acid oxidation at the same time.

3) The authors should at least discuss the possibility that Drp1 may affect other organelles, e.g., peroxisomes, that are involved in fatty acid oxidation.

4) The experiments in Figure 5, Supplementary Figure 2 and Figure 5—figure supplement 1, reveal significantly less AgRP staining in cell bodies and projections in the Drp1 cKO mice compared to controls and significantly more aMSH in cell bodies and projections. Is that because there is less *Agrp* mRNA and more *Pomc* mRNA in the cKO mice?

5) AgRP neurons are not a uniform population; there differences along rostral-caudal region, differences in gene expression and they projection axons to specific brain regions. The authors should take this into consideration when discussing their data or demonstrate uniformity of effect across the rostral-caudal axis.

Included below are the original reviews that provide further insight into issues that the reviewers considered need attention.

*Reviewer #1:*

AgRP neuron activation, which normally occurs in response to food deprivation, enhances food intake and decreases energy expenditure. So, reduced AgRP neuron activity is predicted to have the opposite effect as shown here. There is some concern that the energy expenditure (EE) was measured at 4 months when the mice were already about 2 g lighter than controls (Figure 6). Thus, expressing EE on a per gram basis (rather than per mouse) may artificially enhance the effect. It is easier to appreciate a difference in EE if measurements are made before there is a difference in body weight. Figure 6I appears to show EE per mouse/day as a function of body weight but no statistics are provided.

The Discussion largely reiterates the Abstract and Results without providing any insight into why mitochondrial fission is important for normal AgRP function. Some discussion of how mitochondrial fission might influence fatty acid metabolism and signaling pathways necessary for normal neuronal function would be appreciated. Does fission allow a more efficient distribution of mitochondria in critical regions of the cell?

Editorial suggestions:

Abstract: Are AgRP neurons crucial if mice can adapt to their loss?

Abstract: Why switch from DRP1 to Drp1 when referring to KO mice?

Abstract: Consider "show decreased fasting-.…" to use consistent past tense in this sentence

Abstract: Consider changing sentence to say "neuronal function and body-weight regulation." Rather than behavior

Introduction: Consider: "Previous studies from our laboratory have shown."

Introduction, last paragraph: Remove "may" because this sentence is already hedged by "suggesting"

Subsection “Fasting induces significant upregulation of *Dnm1l* mRNA in AgRP neurons”: Most authors use Cre rather than cre, as in Figure 3—figure supplement 1.

Subsection “Fasting induces significant upregulation of *Dnm1l* mRNA in AgRP neurons”: The labeling on the figure should say *Dnm1l*/*Actb*.

Subsection “Fasting induces significant activation of DRP1 protein in AgRP neurons”: Consider: "We found that the percent of.…was significantly increased"

Subsection “Fasting triggers mitochondrial β-oxidation in the hypothalamic neuro”: Consider: "hypothalamic neurons respond to…"

Subsection “Deletion of *Dnm1l* in AgRP neurons attenuates mitochondrial functions”: Define FA

Subsection “Deletion of *Dnm1l* in AgRP neurons attenuates mitochondrial functions”: “No difference in.….. rate was observed in." 2.5 mM and 0.5 mM (always leave spaces between numbers and units)

Subsection “Inducible and selective deletion of *Dnm1l* in AgRP neurons decreases neuronal activation and projection of AgRP neurons in the hypothalamus” (and elsewhere including figures): There is only one Fos gene in mouse so no need to distinguish cellular c-Fos from viral v-Fos.

Subsection “Deletion of *Dnm1l* in AgRP neurons alters energy metabolism”: Consider "was observed 3 weeks after the start of TMX treatment."

Discussion, end of first paragraph: It seems odd to include "innate" in this sentence because the word usually refers to behavior, not intracellular processes. It is not clear what "every 3 days with 5 times fasting" means. Every 3 days for 5 times is clear enough, but why fasting?

Subsection “Ribotag assays”, last paragraph and Figure 1G-K legend: Gene name should be *Dnm1l* (italics)

Subsection “Ribotag assays”, last paragraph: Gene names should be Actb (Italics)

Subsection “Hypothalamic primary neuronal cell culture: Consider "In brief, we carefully removed the hypothalamus and placed it in a culture dish containing a small volume of.…." Out of curiosity, what fraction of cells express tdTomato after this TMX treatment?

Subsection “Ghrelin administration”: "Individually housed 4-month-old mice.…"

Subsection “Electron microscopy analysis”: "Mice (5 months old) were…" To avoid starting a sentence with a number

Subsection “Measurement of circulating hormones” and throughout the figure legends: "Five-month-old mice.…" or "4-month-old"

Figure 1 legend: "3-month-old fed or fasted mice" add hyphens

Figure 6 legend: 4-month-old (add hyphens, singular)

Figure 3—figure supplement 1 legend: "AgRP-neuron-specific, *Dnm1l* (italics) deleted mice"

*Reviewer #2:*

In the manuscript entitled "DRP1 is required for AgRP neuronal activity and feeding", Jin and co-authors provide evidence that, in mice, DRP1 in AgRP neurons plays an important role in the regulation of AgRP activation thereby contributing to the control of whole-body energy homeostasis. The authors show that during fasting, expression of phosphorylated, active DRP1 is increased in AgRP neurons resulting in the fragmentation of the mitochondrial network. However, upon cell type specific DRP1 deletion AgRP neurons are protected from fasting-induced fragmentation. In addition, DRP1 KO neurons present decreased neuronal activity in the fasted state as well as upon ghrelin treatment, concomitant with reduced AgRP-reactive projections within the hypothalamus. The metabolic consequences caused by these alterations lead to decreased fatty acids oxidation, reduced body weight, fat mass and food intake as well as increased energy expenditure. Given the broad interest in understanding how the CNS controls energy and glucose metabolism, the data are relevant and contribute to a better understanding of the mechanisms implicated in this regulation. The manuscript is very well written and the data largely support the authors' conclusions.

However, a few points should be addressed to improve the current manuscript:

In Figure 2, authors specify that primary hypothalamic neurons were used for measuring β-oxidation upon treatment with different glucose concentrations. However, no data are provided for what contribution AgRP neurons make to these cultures. Hypothalamic neuronal culture purity should be verified via qPCR, western-blot or ICC to determine the proportion of AgRP neurons in the culture.

Results in Figure 3 show that mitochondrial respiration is affected in primary hypothalamic neurons upon AgRP-neuron restricted DRP1 deletion. Nevertheless, the authors do not provide any knock-out confirmation and efficiency for this in vitro model. Knock-out validation and knock-out efficiency should be provided. Moreover, in the Materials and methods section it is described that primary neurons were treated with 4-hydroxytamoxifen to induce the CreERT2-dependent deletion. However, the authors do not include any control group in the seahorse data. They should include a control group of primary neurons treated with 4-hydroxytamoxifen to probe that the treatment itself does not have any detrimental effects for these primary cells.

In Figure 5, Supplementary Figure 2 and Figure 5—figure supplement 1, the authors investigate AgRP- and POMC- immunoreactive projections to several nuclei within the hypothalamus. However, the authors provide no controls to assess whether potentially changes in AgRP or α-MSH expression contribute to the changes in the projection density. Here, the reported reduction of AgRP fibers or the increase of α-MSH fibers can be caused, indeed, by an alteration in the innervation or because the expression of AgRP and α-MSH are reduced or increased, respectively. Therefore, the authors should test whether the expression of AgRP and POMC are changed between controls and DRP1 KO animals.

Finally, in Figure 7, IHC experiments where AgRP neurons are activated upon ghrelin treatment in control and DRP1 KO animals are shown. However, no baseline activation upon saline treatment is provided. This would be important to compare the activation upon Ghrelin treatment.

*Reviewer #3:*

The alternative hypothesis is that lack of DRP1 may impair peroxisome fission/function, and thus b-oxidation of fatty acids, a central function of peroxisomes, is not investigated or discussed in the paper. For this paper to be accepted, experiments either ruling out this possibility or adding it as an additional mechanism are necessary.

In the paper, the authors make two claims related to AgRP neurons: 1) that mitochondrial dynamics are important for their function, and 2) so is fatty acid oxidation. However, the assays presented in Figures 2 and 3 related to fatty acid oxidation use primary hypothalamic cell culture. As AgRP neurons make up a small fraction of the hypothalamus, claims in this regard should be tempered to prevent misinterpretation by the reader, and the two data sets should be presented together. Alternatively, additional testing using an *Agrp* neuron-specific or enriched primary culture would be acceptable.

For Figure 1G, although there is a clear (and expected) difference of *Agrp* transcript in your IP sample, I would have expected to see similar differences in your input sample. However, the y-axis range may be hiding these changes. Please split the y-axis so that we can see any differences in input samples for the *Agrp* transcript. Also, as many publications have found significant and reproducible differences in *Agrp* transcript without using the RiboTag approach, I would have expected the *Agrp* input samples' differences to be significant, even if we cannot see those differences as currently presented. As this data point serves as validation for the methodology of multiple parts of Figure 1, the lack of significance must be explained. If the authors used the same dissection methodology for the primary hypothalamic neurons culture assays (Figures 2 and 3), multiple assays in this paper are in question.

Although the histological images presented in Figures 1, 4, 5, 7, Figure 3—figure supplement 1, Supplementary Figure 2, Figure 5—figure supplement 1 look reasonably convincing, AgRP neurons are spread rostral to caudal within the arcuate nucleus for nearly 2 mm, and conclusions based upon the sample size of 1 histological image out of the entirety of the arc can be misleading. Full stereology of the arcuate nucleus (at least six sections), and the PVN (at least four sections), is necessary to support these claims.

---

## [Author Response]

Essential revisions:1) The authors should provide more compelling evidence that energy expenditure (EE) is increased in mice lacking Drp1 in AgRP neurons because measuring EE per gram body weight when the weights of the mice differs can be misleading.

Energy expenditure (EE) has been recalculated using the lean mass that was not different between control and knockout mice.

2) For the data in Figure 2, it is not clear to what extent AgRP neurons contribute to the difference fatty acid oxidation. Comparison of the data in Figure 2 (cultures from wild-type mice) with the data in Figure 3 (Drp1 cKO mice) are used to demonstrate the importance of Drp1 for fatty acid oxidation. This comparison is troublesome because they were done with primary cultures prepared and assayed at different times and the extent of Drp1 KO was not established. A better design would be to prepare cultures from the Agrp-CreERt::Dnml1(fl/fl) mice that were either treated with tamoxifen or not and then assayed for fatty acid oxidation at the same time.

As suggested, we have now added in Figure 3 data from *Agrp-CreERt::Dnml1^fl/fl^* mice treated with vehicle as control group. Our conclusion stands that DRP1 plays a role in PA oxidation.

3) The authors should at least discuss the possibility that Drp1 may affect other organelles, e.g., peroxisomes, that are involved in fatty acid oxidation.

As suggested, we have now added a paragraph in the Discussion addressing this possibility:

“In addition to mitochondria, DRP1 has been also shown to enable peroxisomal fission. Peroxisomes are single-membrane organelles that similar to mitochondria catalyze the breakdown of long chain fatty acids through beta-oxidation and regulate the maintenance of redox homeostasis (Smith and Aitchison, 2013). Because of these shared properties and metabolic pathways, we cannot exclude a possible involvement of peroxisomes in the phenotype observed in our mice. Further studies are warranted to address this issue.”

4) The experiments in Figure 5, Supplementary Figure 2 and Figure 5—figure supplement 1, reveal significantly less AgRP staining in cell bodies and projections in the Drp1 cKO mice compared to controls and significantly more aMSH in cell bodies and projections. Is that because there is less Agrp mRNA and more Pomc mRNA in the cKO mice?

We have now performed and added qPCR data from RiboTag mice (Figure 5—figure supplement 2B and C) and found no significant differences in AgRP and POMC mRNA levels between Drp1 cKO and control mice.

5) AgRP neurons are not a uniform population; there differences along rostral-caudal region, differences in gene expression and they projection axons to specific brain regions. The authors should take this into consideration when discussing their data or demonstrate uniformity of effect across the rostral-caudal axis.

We have now added data in Figure 4, Figure 4—figure supplement 1 and Figure 5 showing % Fos positive-AgRP neurons and AgRP projections to the PVN in different levels of the hypothalamus. We found significant differences at every level examined across the rostral-caudal axis of the hypothalamus.

Included below are the original reviews that provide further insight into issues that the reviewers considered need attention.Reviewer #1:AgRP neuron activation, which normally occurs in response to food deprivation, enhances food intake and decreases energy expenditure. So, reduced AgRP neuron activity is predicted to have the opposite effect as shown here. There is some concern that the energy expenditure (EE) was measured at 4 months when the mice were already about 2 g lighter than controls (Figure 6). Thus, expressing EE on a per gram basis (rather than per mouse) may artificially enhance the effect. It is easier to appreciate a difference in EE if measurements are made before there is a difference in body weight. Figure 6I appears to show EE per mouse/day as a function of body weight but no statistics are provided.

Energy expenditure (EE) has been recalculated using the lean mass that was not different between control and knockout mice.

The Discussion largely reiterates the Abstract and Results without providing any insight into why mitochondrial fission is important for normal AgRP function. Some discussion of how mitochondrial fission might influence fatty acid metabolism and signaling pathways necessary for normal neuronal function would be appreciated. Does fission allow a more efficient distribution of mitochondria in critical regions of the cell?

We have now added an additional paragraph discussing the potential role of mitochondrial dynamics in mitochondrial subcellular distribution:

“In addition to these functions, changes in shape and size of mitochondria may also affect the ability of a cell to distribute its mitochondrial population to specific subcellular locations. This function is especially important in highly polarized cells, such as neurons. Future studies assessing mitochondrial dynamics with changes in mitochondrial subcellular distribution will address this point.”

Editorial suggestions:Abstract: Are AgRP neurons crucial if mice can adapt to their loss?

One could argue that they are so crucial that the brain has developed a mechanism to adapt to their loss.

Abstract: Why switch from DRP1 to Drp1 when referring to KO mice?

To indicate protein. However, we change it to Drp1 has we interfered with the gene.

Abstract: Consider "show decreased fasting-.…" to use consistent past tense in this sentence

Corrected as suggested

Abstract: Consider changing sentence to say "neuronal function and body-weight regulation." Rather than behavior

Corrected as suggested

Introduction: Consider: "Previous studies from our laboratory have shown."

Corrected as suggested

Introduction, last paragraph: Remove "may" because this sentence is already hedged by "suggesting"

Corrected as suggested

Subsection “Fasting induces significant upregulation of Dnm1l mRNA in AgRP neurons”: Most authors use Cre rather than cre, as in Figure 3—figure supplement 1.

Corrected as suggested

Subsection “Fasting induces significant upregulation of Dnm1l mRNA in AgRP neurons”: The labeling on the figure should say Dnm1l/Actb.

Corrected as suggested

Subsection “Fasting induces significant activation of DRP1 protein in AgRP neurons”: Consider: "We found that the percent of.…was significantly increased"

Corrected as suggested

Subsection “Fasting triggers mitochondrial β-oxidation in the hypothalamic neuro”: Consider: "hypothalamic neurons respond to…"

Corrected as suggested

Subsection “Deletion of Dnm1l in AgRP neurons attenuates mitochondrial functions”: Define FA

We meant PA (Palmitic acid). FA has been now corrected to PA

Subsection “Deletion of Dnm1l in AgRP neurons attenuates mitochondrial functions”: “No difference in.….. rate was observed in..." 2.5 mM and 0.5 mM (always leave spaces between numbers and units)

Corrected as suggested

Subsection “Inducible and selective deletion of Dnm1l in AgRP neurons decreases neuronal activation and projection of AgRP neurons in the hypothalamus” (and elsewhere including figures): There is only one Fos gene in mouse so no need to distinguish cellular c-Fos from viral v-Fos.

Corrected to Fos now throughout the manuscript and figures.

Subsection “Deletion of Dnm1l in AgRP neurons alters energy metabolism”: Consider "was observed 3 weeks after the start of TMX treatment."

Corrected as suggested.

Discussion, end of first paragraph: It seems odd to include "innate" in this sentence because the word usually refers to behavior, not intracellular processes.

“innate” has been removed.

It is not clear what "every 3 days with 5 times fasting" means. Every 3 days for 5 times is clear enough, but why fasting?

We observed that tamoxifen-induced tdTomato expression selectively in AgRP neurons was not sufficient without fasting. We believe that fasting by increasing AgRP transcription and neuronal activity, promotes increased Cre recombinase induction.

Subsection “Ribotag assays”, last paragraph and Figure 1G-K legend: Gene name should be Dnm1l (italics)

Corrected as suggested

Subsection “Ribotag assays”, last paragraph: Gene names should be Actb (Italics)

Corrected as suggested

Subsection “Hypothalamic primary neuronal cell culture: Consider "In brief, we carefully removed the hypothalamus and placed it in a culture dish containing a small volume of.…."Out of curiosity, what fraction of cells express tdTomato after this TMX treatment?

These data are now shown in Figure 2—figure supplement 1A-D

Subsection “Ghrelin administration”: "Individually housed 4-month-old mice.…"

Corrected as suggested

Subsection “Electron microscopy analysis”: "Mice (5 months old) were…" To avoid starting a sentence with a number

Corrected as suggested

Subsection “Measurement of circulating hormones” and throughout the figure legends: "Five-month-old mice.…" or "4-month-old"

Corrected as suggested

Figure 1 legend: "3-month-old fed or fasted mice" add hyphens

Corrected as suggested

Figure 6 legend: 4-month-old (add hyphens, singular)

Corrected as suggested

Figure 3—figure supplement 1 legend: "AgRP-neuron-specific, Dnm1l (italics) deleted mice"

Corrected as suggested

Reviewer #2:In the manuscript entitled "DRP1 is required for AgRP neuronal activity and feeding", Jin and co-authors provide evidence that, in mice, DRP1 in AgRP neurons plays an important role in the regulation of AgRP activation thereby contributing to the control of whole-body energy homeostasis. The authors show that during fasting, expression of phosphorylated, active DRP1 is increased in AgRP neurons resulting in the fragmentation of the mitochondrial network. However, upon cell type specific DRP1 deletion AgRP neurons are protected from fasting-induced fragmentation. In addition, DRP1 KO neurons present decreased neuronal activity in the fasted state as well as upon ghrelin treatment, concomitant with reduced AgRP-reactive projections within the hypothalamus. The metabolic consequences caused by these alterations lead to decreased fatty acids oxidation, reduced body weight, fat mass and food intake as well as increased energy expenditure. Given the broad interest in understanding how the CNS controls energy and glucose metabolism, the data are relevant and contribute to a better understanding of the mechanisms implicated in this regulation. The manuscript is very well written and the data largely support the authors' conclusions.However, a few points should be addressed to improve the current manuscript:In Figure 2, authors specify that primary hypothalamic neurons were used for measuring β-oxidation upon treatment with different glucose concentrations. However, no data are provided for what contribution AgRP neurons make to these cultures. Hypothalamic neuronal culture purity should be verified via qPCR, western-blot or ICC to determine the proportion of AgRP neurons in the culture.

These data are now shown in Figure 2—figure supplement 1A-D.

Results in Figure 3 show that mitochondrial respiration is affected in primary hypothalamic neurons upon AgRP-neuron restricted DRP1 deletion. Nevertheless, the authors do not provide any knock-out confirmation and efficiency for this in vitro model. Knock-out validation and knock-out efficiency should be provided.

These data are now shown in Figure 3—figure supplement 2F.

Moreover, in the Materials and methods section it is described that primary neurons were treated with 4-hydroxytamoxifen to induce the CreERT2-dependent deletion. However, the authors do not include any control group in the seahorse data.

As suggested by this reviewer, we have now added in Figure 3G-J data from *Agrp-CreERT2::Dnml1(fl/fl)* mice treated with vehicle as control group. Our conclusion stands that DRP1 plays a role in FA oxidation.

They should include a control group of primary neurons treated with 4-hydroxytamoxifen to probe that the treatment itself does not have any detrimental effects for these primary cells.

We have now added these in Figure 2—figure supplement 1E and Figure 3—figure supplement 2E.

In Figure 5, Supplementary Figure 2 and Figure 5—figure supplement 1, the authors investigate AgRP- and POMC- immunoreactive projections to several nuclei within the hypothalamus. However, the authors provide no controls to assess whether potentially changes in AgRP or α-MSH expression contribute to the changes in the projection density. Here, the reported reduction of AgRP fibers or the increase of α-MSH fibers can be caused, indeed, by an alteration in the innervation or because the expression of AgRP and α-MSH are reduced or increased, respectively. Therefore, the authors should test whether the expression of AgRP and POMC are changed between controls and DRP1 KO animals.

We have now performed qPCR analysis from RiboTag mice (Figure 5—figure supplement 2B and C) and found no significant differences in AgRP and POMC mRNA levels between Drp1 cKO and control mice.

Finally, in Figure 7, IHC experiments where AgRP neurons are activated upon ghrelin treatment in control and DRP1 KO animals are shown. However, no baseline activation upon saline treatment is provided. This would be important to compare the activation upon Ghrelin treatment.

In the present study, ghrelin was administrated in overnight fed mice at 9AM. Saline administration was not performed as in fed state Fos expression in AgRP neurons in very low if detected.

Reviewer #3:The alternative hypothesis is that lack of DRP1 may impair peroxisome fission/function, and thus b-oxidation of fatty acids, a central function of peroxisomes, is not investigated or discussed in the paper. For this paper to be accepted, experiments either ruling out this possibility or adding it as an additional mechanism are necessary.

We have now added a paragraph in the Discussion on this possibility:

“In addition to mitochondria, DRP1 has been also shown to enable peroxisomal fission. Peroxisomes are single-membrane organelles that similar to mitochondria catalyze the breakdown of long chain fatty acids through beta-oxidation and regulate the maintenance of redox homeostasis (Smith and Aitchison, 2013). Because of these shared properties and metabolic pathways, we cannot exclude a possible involvement of peroxisomes in the phenotype observed in our mice. Further studies are warranted to address this issue.”

In the paper, the authors make two claims related to AgRP neurons: 1) that mitochondrial dynamics are important for their function, and 2) so is fatty acid oxidation. However, the assays presented in Figures 2 and 3 related to fatty acid oxidation use primary hypothalamic cell culture. As AgRP neurons make up a small fraction of the hypothalamus, claims in this regard should be tempered to prevent misinterpretation by the reader, and the two data sets should be presented together. Alternatively, additional testing using an Agrp neuron-specific or enriched primary culture would be acceptable.

As also suggested by reviewer 2, we have now added in Figure 3 data from primary cultures treated with vehicle (as control group) derived from *Dnm1l^fl/fl^; Agrp^Cre:ERT2^*;tdTomato mice. In addition, we have added data in Figure 2—figure supplement 1 and Figure 3—figure supplement 2 data showing that the 2 control cultures behave similarly.

For Figure 1G, although there is a clear (and expected) difference of Agrp transcript in your IP sample, I would have expected to see similar differences in your input sample. However, the y-axis range may be hiding these changes. Please split the y-axis so that we can see any differences in input samples for the Agrp transcript. Also, as many publications have found significant and reproducible differences in Agrp transcript without using the RiboTag approach, I would have expected the Agrp input samples' differences to be significant, even if we cannot see those differences as currently presented.

We have now corrected the y axis to see the difference in input samples in Figure 1G and H.

As this data point serves as validation for the methodology of multiple parts of Figure 1, the lack of significance must be explained. If the authors used the same dissection methodology for the primary hypothalamic neurons culture assays (Figures 2 and 3), multiple assays in this paper are in question.

We do see significant difference in the input samples which are now visible after changing the y axis. Furthermore, we did use different methods for dissection in the RiboTag experiment versus the primary hypothalamic neuronal culture experiments. We collected the arcuate nucleus of adult mice under a stereomicroscope for the RiboTag experiments, while for the primary hypothalamic neuronal culture assays, we collected the MBH (which did not contain only the arcuate) of pups.

Although the histological images presented in Figures 1, 4, 5, 7, Figure 3—figure supplement 1, Supplementary Figure 2 and Figure 5—figure supplement 1 look reasonably convincing, AgRP neurons are spread rostral to caudal within the arcuate nucleus for nearly 2 mm, and conclusions based upon the sample size of 1 histological image out of the entirety of the arc can be misleading. Full stereology of the arcuate nucleus (at least six sections), and the PVN (at least four sections), is necessary to support these claims.

We have now revised our analyses using sections at different levels of the rostral- caudal axis. See Figure 4, Figure 4—figure supplement 1 and Figure 5.